# HybridFlow: Resource-Adaptive Subtask Routing for Efficient Edge-Cloud LLM Inference

**Jiangwen Dong** [1]   **Jiayu Li** [1]   **Tianhang Zheng** [2]   **Wanyu Lin** [1]

## Abstract

Edge-cloud collaborative inference is crucial for LLM-powered edge devices, as on-device models often lack the required reasoning capability, while cloud-only inference can be costly and slow under strict latency and token/API budgets. However, existing edge-cloud collaboration methods typically route input tasks based on their estimated difficulty. These static, coarse heuristics overlook subtask dependencies, missing opportunities for parallel execution and budget-adaptive routing. To this end, we propose **HybridFlow**, a resource-adaptive edge-cloud inference framework that enables parallel execution of interdependent subtasks. Specifically, we build a dependency-aware DAG for each input task, facilitating concurrent execution of subtasks once their dependencies are resolved, thereby reducing end-to-end latency. Additionally, we propose a dynamic benefit–cost utility model, optimizing the trade-off between accuracy, token/API cost, and latency in real-time. This dynamic routing minimizes unnecessary cloud usage while preserving reasoning quality. Across GPQA, MMLU-Pro, AIME24, and LiveBench-Reasoning, HybridFlow improves the cost-accuracy trade-off, reducing latency and cloud API usage while maintaining competitive accuracy. Code: https://github.com/WanyuGroup/ICML2026_HybridFlow

## 1. Introduction

Large language models (LLMs) have recently demonstrated remarkable capabilities across a wide range of tasks, especially those requiring multi-step reasoning, complex decision-making, and problem solving (Guo et al., 2025; Lin et al., 2025; Xiong et al., 2024; Yu et al., 2025). Yet, these gains come with substantial practical costs: high inference latency, large memory footprints, and non-trivial API expenses when served from the cloud (Yuan et al., 2025; Jiang et al., 2025). Such costs are particularly prohibitive for latency-sensitive and resource-constrained edge devices, where the goal is to achieve acceptable accuracy under strict latency and cost budgets, rather than maximizing accuracy in isolation (Ye et al., 2025; Tian et al., 2025).

A natural alternative is to deploy small models (SMs) on-device, benefiting from lightweight designs such as quantization and distillation (Wang et al., 2025a; Qu et al., 2025). However, SM-only solutions often struggle on tasks that demand deep reasoning or broad knowledge due to limited model capacity. On the other hand, cloud-only LLM inference often conflicts with practical deployment constraints, particularly API cost budgets; end-to-end latency can also become prohibitive under unfavorable network or service conditions. This tension motivates edge-cloud collaboration, where an on-device SM collaborates with a cloud LLM to balance accuracy, latency, and cost (Yuan et al., 2025; Akhauri et al., 2025).

Despite its promise, existing edge-cloud collaboration methods typically make coarse-grained decisions—e.g., routing at the query level or at a fixed reasoning-step granularity—based mainly on predicted task difficulty (Ding et al., 2024; Shao et al., 2025; Chen et al., 2024a). Two limitations follow. First, coarse routing may miss opportunities for fine-grained parallelism: many complex queries naturally decompose into interdependent parts, where unlocked parts could be executed concurrently, but coarse allocation forces largely sequential execution. Second, practical deployments face time-varying budgets and conditions (e.g., fluctuating network latency, dynamic API budgets, and varying edge load), while prior approaches often use static heuristics or fixed thresholds that do not adapt online. These gaps raise the following question: *How can we design an edge-cloud framework that performs adaptive, budget-aware scheduling and routing at a fine granularity, enabling fast and API-efficient inference on complex reasoning tasks?*

To this end, we propose HybridFlow, a utility-driven edge-

---

[1]Department of Computing, The Hong Kong Polytechnic University, Hong Kong SAR, China [2]State Key Lab. of Blockchain and Data Security, Zhejiang University, China. Correspondence to: Wanyu Lin <wan-yu.lin@polyu.edu.hk>.

*Proceedings of the 43rd International Conference on Machine Learning*, Seoul, South Korea. PMLR 306, 2026. Copyright 2026 by the author(s).

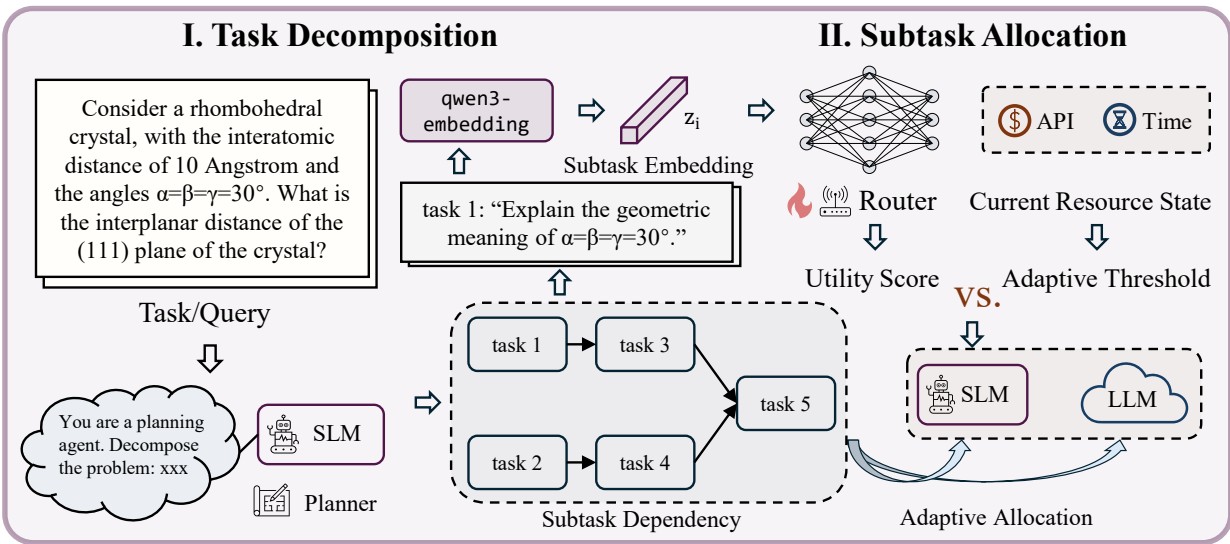

*Figure 1.* Overview of the **HybridFlow** framework. (I) Task Decomposition: The planner decomposes a complex query into a directed acyclic graph of subtasks with explicit dependencies. (II) Subtask Allocation: The router encodes each subtask with semantic and resource features, predicts its utility score considering quality, latency, and API cost, and adaptively allocates it to either the edge SLM or the cloud LLM for efficient collaboration.

cloud inference framework that performs subtask-level, dependency-aware collaboration. HybridFlow first decomposes a complex query into a set of subtasks with explicit dependencies, represented as a directed acyclic graph (DAG). Then, as dependencies are resolved, HybridFlow schedules newly unlocked subtasks and routes each subtask online to either the edge SM or the cloud LLM using a learned benefit-cost utility predictor, combined with an adaptive budget-aware decision rule. This design explicitly couples (i) dependency-aware decomposition, (ii) parallel scheduling, and (iii) online budget-aware routing, which allows HybridFlow to reduce end-to-end latency and cloud API usage under tight resource constraints.

In summary, our contributions are:

- **Fine-grained, dependency-aware edge-cloud inference.** We introduce HybridFlow, a subtask-level edge-cloud framework that represents complex reasoning as a DAG and executes subtasks in a dependency-triggered manner to expose parallelism beyond query-level routing.

- **Online budget-aware routing with utility modeling.** We develop a resource-aware routing mechanism that uses a learned benefit-cost utility model and an adaptive decision rule to allocate each subtask to edge or cloud under latency and API budgets.

- **Empirical evidence on challenging reasoning benchmarks.** We evaluate HybridFlow on four challenging benchmarks (GPQA, AIME24, LiveBench-Reasoning, and MMLU-Pro), showing improved latency-cost

trade-offs while maintaining competitive accuracy against strong structured reasoning baselines.

## 2. Related Work

**Decomposed and Dependency-Aware Reasoning.** A growing body of research aims to improve LLM reasoning efficiency by decomposing inference into smaller, more manageable units. CoT prompting (Wei et al., 2022) enhances reasoning accuracy but often increases API and latency costs due to long intermediate chains. More recent works have introduced structured decomposition and partial parallelism to address these issues. For example, SoT (Ning et al., 2024) and PASTA (Jin et al., 2025) elicit intermediate sub-questions or steps that can be executed concurrently, optimizing both accuracy and efficiency. In contrast to linear decompositions, ToT (Yao et al., 2023) organizes reasoning into a tree structure, allowing deliberate search over candidate thoughts, while GoT (Besta et al., 2024) and S-DAG (Dong et al., 2026) extend this idea to graph-based reasoning, enabling flexible, reusable intermediate nodes with complex dependency patterns.

**Budgeted Edge-Cloud Routing and Online Adaptation.** Edge-cloud collaboration often involves routing mechanisms where simpler models handle easier tasks and offload more complex ones to stronger models, or where tasks are decomposed and distributed across multiple models (Wang et al., 2025b; Siyan et al., 2025; Ding et al., 2024; Shao et al., 2025). However, many systems operate at coarse granularity (e.g., query-level or stage-level) and do not explicitly opti-

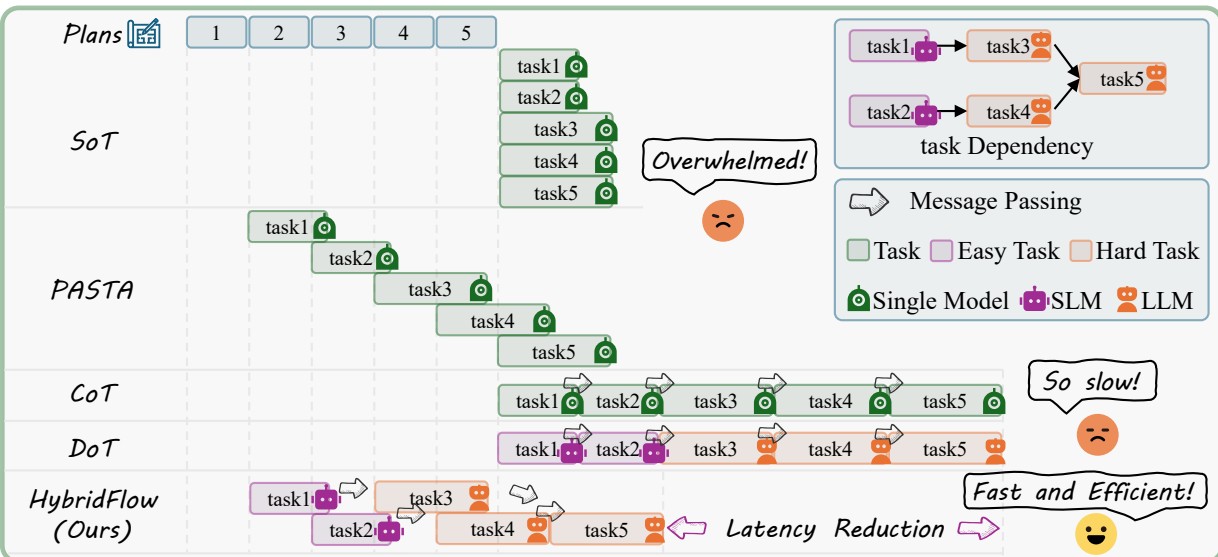

*Figure 2.* Overview of comparative LLM inference pipeline. **HybridFlow** uniquely integrates dependency-aware planning concurrently with parallel execution. This achieves an optimal balance between speed and reasoning quality by actively exploiting concurrent opportunities within a logically constrained workflow.

mize for tight, multi-resource constraints such as latency and API budgets. Cost-aware routing methods, like LLM cascades using mixture-of-thought representations (Yue et al., 2024) and FrugalGPT (Chen et al., 2024b), address budget-quality trade-offs by selectively invoking stronger models at various stages. Related work like SplitReason (Akhauri et al., 2025) also explores fine-grained offloading, learning to delegate specific reasoning steps to stronger models. However, many of these learned routers rely on stationary assumptions or fixed thresholds, which can hinder performance when the workload, network conditions, or pricing fluctuate. In contrast, HybridFlow adapts dynamically to these changes, optimizing edge-cloud allocation in real-time for more robust and efficient reasoning.

# 3. HybridFlow Framework

HybridFlow enables efficient edge-cloud collaborative inference by combining dependency-aware task planning with cost-aware routing. As illustrated in Fig. 1, the system has two tightly connected components. First, an edge-side planner decomposes a complex query into a set of interdependent subtasks and constructs a task-level DAG, enabling each subtask to execute as soon as its prerequisites are resolved. This exposes fine-grained parallelism and reduces wall-clock latency compared with strictly sequential reasoning. Second, a resource-aware router assigns each subtask to either the edge model $M_{\text{edge}}$ or the cloud LLM $M_{\text{cloud}}$ based on its predicted utility and the system's real-time budget status. By coupling decomposition with budget-constrained routing, HybridFlow explicitly balances accuracy and efficiency.

We summarize the notations in Table 5 and outline the full procedure in Algorithm 1.

## 3.1. Preliminaries

We consider an edge-cloud inference setting in which a small edge model $M_{\text{edge}}$ collaborates with a large cloud model $M_{\text{cloud}}$ to solve a query $Q$. The query is decomposed into $n$ subtasks $\{t_1, \ldots, t_n\}$, each executable either on $M_{\text{edge}}$ or on $M_{\text{cloud}}$. The routing decision for each subtask must trade off the accuracy benefit of using the cloud against its resource cost.

For each subtask $t_i$, we define:

- $\Delta q_i$: expected *accuracy gain* of executing $t_i$ on the cloud instead of the edge;

- $\Delta l_i$: additional *latency cost* (in seconds) incurred by cloud execution;

- $\Delta k_i$: additional *API usage cost* (in price units) incurred by cloud execution.

In practice, $\Delta q_i$ is obtained via an offline outcome-based credit assignment procedure. For each subtask $i$, we construct paired executions that differ only in whether subtask $i$ is processed on the cloud or on the edge while keeping the rest of the pipeline unchanged.

**Definition 3.1 (Normalized Cost).** To place latency and API usage on a common scale, we define a normalized offloading cost. For each subtask $t_i$, the normalized cost of

**Algorithm 1** Dependency-Aware Subtask Scheduling with Budget-Adaptive Routing

---

**Require:** User query $Q$
**Ensure:** Final response $R$
  **Stage 1: Task Decomposition and DAG construction**
  $(T, E) \leftarrow \text{Decompose}(Q; M_P)$ {subtasks and dependencies}
  $G \leftarrow \text{ValidateAndRepair}(T, E)$ {fallback to chain if invalid}
  Initialize $\deg(\cdot)$ as in-degree in $G$
  Initialize frontier queue $\mathcal{Q} \leftarrow \{x \in T : \deg(x) = 0\}$
  **Stage 2: Budget-Adaptive Routing and Execution**
  Initialize $C_{\text{used}}(t) \leftarrow 0$
  **while** $\mathcal{Q}$ not empty **do**
    Pop a ready subtask $x$ from $\mathcal{Q}$.
    Obtain subtask embedding: $z(x) = \text{embedding}(x)$
    Predict utility score $\hat{u}(x) = \sigma\big(f_\theta(z(x), C_{\text{used}}(t))\big)$.
    Compute adaptive threshold $\tau_t$ from current resource usage.
    **if** $\hat{u}(x) > \tau_t$ **then**
      $R_x \leftarrow M_{\text{cloud}}(x)$ {Cloud execution}
      Update $C_{\text{used}}(t)$.
    **else**
      $R_x \leftarrow M_{\text{edge}}(x)$ {Edge execution}
      Update $C_{\text{used}}(t)$.
    **end if**
    Append $R_x$ to global context and push newly unblocked subtasks into $\mathcal{Q}$.
  **end while**
  **Stage 3: Final Aggregation**
  Combine all sub-results in topological order to form $R$.
  **return** $R$

---

offloading is

$$c_i = \text{clip}\left(\left(\frac{\Delta l_i}{l_{\max}^{\text{sub}}} + \frac{\Delta k_i}{k_{\max}^{\text{sub}}}\right)/2, \, 0, \, 1\right) \in [0,1], \quad (1)$$

where $l_{\max}^{\text{sub}}$ and $k_{\max}^{\text{sub}}$ are per-subtask upper bounds on additional latency and API cost. Typically $\Delta l_i \leq l_{\max}^{\text{sub}}$ and $\Delta k_i \leq k_{\max}^{\text{sub}}$, so $c_i$ is bounded and comparable across subtasks. We use a binary variable $r_i \in \{0,1\}$ to indicate whether $t_i$ is offloaded to the cloud ($r_i = 1$) or executed on the edge ($r_i = 0$). Intuitively, subtasks with larger $\Delta q_i$ and smaller $c_i$ are more attractive to offload.

**Definition 3.2** (Utility)**.** The *utility* of offloading subtask $t_i$ is the normalized benefit-cost ratio

$$u_i = \text{clip}\left(\frac{\Delta q_i}{c_i + \varepsilon}, \, 0, \, 1\right), \quad (2)$$

where $\varepsilon > 0$ is a small constant (e.g., $10^{-4}$) for numerical stability and $\text{clip}(\cdot, 0, 1)$ truncates the value to $[0,1]$. Conceptually, $u_i$ measures the accuracy improvement per unit normalized cost and serves as the ideal offloading score.

**Knapsack Formulation.** Using these quantities, subtask allocation can be written as a resource-constrained optimization problem. Let $\mathbf{r} = (r_1, \ldots, r_n)$ denote the allocation vector. We seek to maximize the total accuracy gain under a normalized resource budget:

$$\max_{\mathbf{r} \in \{0,1\}^n} \sum_{i=1}^n r_i \, \Delta q_i \quad \text{s.t.} \quad \sum_{i=1}^n r_i \, c_i \leq C_{\max}, \quad (3)$$

where $C_{\max}$ is the total normalized budget per query. This is exactly a 0-1 knapsack problem: each subtask is an item with value $\Delta q_i$ and weight $c_i$ under capacity $C_{\max}$.

**Lagrangian Relaxation and Threshold Structure.** Introducing a Lagrange multiplier $\lambda \geq 0$ for the budget constraint, the Lagrangian is

$$\mathcal{L}(\mathbf{r}, \lambda) = \sum_{i=1}^n r_i \Delta q_i \, - \, \lambda\left(\sum_{i=1}^n r_i c_i - C_{\max}\right) \quad (4)$$

$$= \lambda C_{\max} + \sum_{i=1}^n r_i(\Delta q_i - \lambda c_i). \quad (5)$$

For fixed $\lambda$, the relaxed problem decouples across subtasks, and the optimal decision follows a threshold rule:

$$r_i^\star(\lambda) = \mathbb{I}\big[\Delta q_i - \lambda c_i > 0\big] = \mathbb{I}\left[\frac{\Delta q_i}{c_i} > \lambda\right]. \quad (6)$$

Thus, an ideal policy offloads subtasks with sufficiently large benefit-cost ratio. This structure motivates HybridFlow's learned utility and online thresholding in Sec. 3.3.

### 3.2. Task Decomposition and Execution Pipeline

Effective task decomposition forms the backbone of HybridFlow, as it determines both the logical structure and the degree of parallelism in the inference pipeline. Given a query $Q$, the planner must identify meaningful subtasks and their dependencies so that reasoning remains coherent while supporting concurrent execution on edge and cloud workers. As illustrated in Fig. 2, existing frameworks typically sit at two extremes: approaches such as SoT (Ning et al., 2024) and PASTA (Jin et al., 2025) aggressively parallelize steps with limited regard for dependencies, while CoT (Wei et al., 2022) and DoT (Shao et al., 2025) enforce strictly sequential execution, preserving correctness but incurring high latency.

HybridFlow sidesteps this dilemma by explicitly modeling dependencies while still exploiting available parallelism. We leverage an edge-deployed planner $M_P$ and elicit its behavior with an Explain-Analyze-Generate (EAG) meta-prompt (Gu et al., 2025). The prompt guides $M_P$ through three stages: (i) identifying key elements of the query, (ii) analyzing and breaking the query into interrelated subtasks, and (iii) generating a structured representation of the overall solution plan. To make this process robust, we curate

high-quality exemplars by evaluating multiple models on 100 queries from s1k (Muennighoff et al., 2025) and selecting examples that exhibit strong logical grounding, clear dependency structure, and non-trivial parallelism. These exemplars are used as few-shot demonstrations for the planner.

Formally, we write the decomposition process as

$$(T, E) \; = \; \mathrm{Decompose}(Q; M_P), \qquad (7)$$

where $T = \{t_1, \ldots, t_n\}$ is the set of subtasks and $E \subseteq T \times T$ encodes directed prerequisite relations. In practice, $M_P$ outputs an XML-formatted plan whose parent fields are parsed into a task-level DAG $G(Q) = (T, E)$. A scheduler maintains a queue of ready subtasks whose parents have completed and dispatches them immediately to either $M_{\mathrm{edge}}$ or $M_{\mathrm{cloud}}$ according to the routing policy in Sec. 3.3. This design preserves explicit dependencies while allowing independent subtasks to proceed in parallel, reducing end-to-end latency compared with purely sequential execution. See prompts in Figure 6.

### 3.3. Utility-based Subtask Routing

The router decides whether to assign a subtask to $M_{\mathrm{cloud}}$ by estimating the utility of offloading to the cloud. It serves as a lightweight approximation to the knapsack allocation in Eq. (3), while adapting its conservativeness online according to real-time budget usage.

**Offline Utility Estimation.** Each subtask $t_i$ is encoded into a semantic embedding $z_i$ using the `qwen3-embedding-0.6b` model (Zhang et al., 2025). A multilayer perceptron (MLP) computes a normalized estimated utility

$$\hat{u}_i = \sigma\big(f_\theta(z_i, C_{\mathrm{used}}(t))\big) \in (0, 1), \qquad (8)$$

where $\sigma$ is the sigmoid function and $C_{\mathrm{used}}(t) = \sum_{j \leq t} r_j c_j$ is the cumulative normalized cost used so far. The estimate $\hat{u}_i$ is trained to approximate the ideal utility $u_i$ in Def. 3.2.

**Router Training.** Supervision is derived by profiling subtasks on both edge and cloud models. The observed accuracy gain $\Delta q_i$ and normalized cost $c_i$ define the utility target $u_i$ via Eq. (2). The router is trained with MSE:

$$\mathcal{L}(\theta) = \frac{1}{N} \sum_{i=1}^{N} \big(\hat{u}_i - u_i\big)^2. \qquad (9)$$

This amortizes offline profiling into a fast online decision and provides a strong warm-start.

**Online Dual Thresholding.** During inference, subtasks become available online as dependencies resolve. Motivated

by the threshold structure in Eq. (6), HybridFlow maintains a dual variable $\lambda_t \geq 0$ as a shadow price for normalized resource consumption and updates it via a projected subgradient step:

$$\lambda_{t+1} = \Big[\lambda_t + \eta\big(C_{\mathrm{used}}(t) - C_{\max}\big)\Big]_+, \qquad (10)$$

where $\eta > 0$ is a step size. When the system overspends, $\lambda_t$ increases and the routing policy becomes more conservative.

We map the shadow price to a normalized routing threshold $\tau_t \in [0, 1]$ via a monotone transform:

$$\tau_t = \mathrm{clip}(\tau_0 + \gamma \lambda_t, \, 0, 1), \qquad (11)$$

where $\tau_0$ is a base threshold and $\gamma$ controls sensitivity. The router then offloads $t_i$ if its (possibly calibrated) utility exceeds $\tau_t$:

$$r_i = \mathbb{I}[\bar{u}_i > \tau_t], \qquad \bar{u}_i = \begin{cases} \tilde{u}_i, & \text{if online calibration,} \\ \hat{u}_i, & \text{otherwise.} \end{cases} \qquad (12)$$

This rule approximates the relaxed Lagrangian policy in Eq. (6) while adapting to real-time budget consumption.

**Contextual Bandit Calibration.** Offline utilities $\hat{u}_i$ may be miscalibrated under system shifts (e.g., cloud latency changes) or task shifts (e.g., different query domains). To adapt online with partial feedback, we introduce a lightweight calibration head that refines $\hat{u}_i$ using runtime context. Let $s_i$ denote a feature vector of online signals, such as remaining budget and planner-provided attributes. We define a calibrated utility:

$$\tilde{u}_i = \mathrm{clip}\big(\alpha \hat{u}_i + \beta + w^\top s_i, \, 0, 1\big), \qquad (13)$$

where $(\alpha, \beta, w)$ are updated online.

HybridFlow only observes the quality gain $\Delta q_i$ when $t_i$ is offloaded ($r_i = 1$), yielding a contextual bandit setting with partial feedback. We define a cost-aware reward consistent with the Lagrangian form:

$$R_i = \Delta q_i - \lambda_t c_i, \qquad (14)$$

and update $(\alpha, \beta, w)$ online using a standard contextual bandit strategy (e.g., LinUCB) to ensure exploration. This calibration is lightweight and can be enabled when robustness to shifts is desired.

## 4. Experiments

### 4.1. Experiment Setup

**Datasets and Metrics.** We evaluate HybridFlow on four reasoning benchmarks spanning mathematical, scientific, and general domains: *GPQA* (Rein et al., 2024);

*Table 1.* Accuracy (%, mean $\pm$ std) of HybridFlow and baseline methods across four benchmarks. **Bold** denotes the highest accuracy and underline indicates the second-highest. Direct Prompt results (shaded) are reference single-model baselines and are excluded from ranking. HybridFlow achieves competitive accuracy while operating under edge-cloud collaboration constraints.

| Method | Model | GPQA | MMLU-Pro | AIME24 | LiveBench-Reasoning | Avg. ($\uparrow$) |
|---|---|---|---|---|---|---|
| Direct Prompt | L3B | $16.89_{\pm1.05}$ | $22.83_{\pm1.31}$ | $4.44_{\pm1.57}$ | $12_{\pm2.86}$ | 14.04 |
| Direct Prompt | G4.1 | $51.79_{\pm1.17}$ | $65.5_{\pm1.47}$ | $37.78_{\pm1.57}$ | $58.25_{\pm0.75}$ | 53.33 |
| CoT (Wei et al., 2022) | L3B | $25.54_{\pm1.57}$ | $31.67_{\pm0.85}$ | $5.56_{\pm1.57}$ | $15.6_{\pm1.93}$ | 19.59 |
| CoT | G4.1 | $\mathbf{57.28_{\pm0.73}}$ | $72_{\pm0.71}$ | $\mathbf{44.42_{\pm1.59}}$ | $\mathbf{62.25_{\pm0.75}}$ | $\mathbf{58.99}$ |
| SoT (Ning et al., 2024) | L3B | $30.24_{\pm0.34}$ | $31.67_{\pm2.87}$ | $1.11_{\pm1.57}$ | $17.33_{\pm1.89}$ | 20.09 |
| SoT | G4.1 | $\underline{56.4_{\pm0.99}}$ | $71.8_{\pm1.03}$ | $28.89_{\pm1.57}$ | $54.5_{\pm1.08}$ | 52.90 |
| PASTA (Jin et al., 2025) | L3B | $28.67_{\pm2.83}$ | $25.84_{\pm2.65}$ | $2.22_{\pm1.92}$ | $14.75_{\pm1.02}$ | 17.87 |
| PASTA | G4.1 | $41.28_{\pm2.87}$ | $\mathbf{75.52_{\pm1.77}}$ | $32.1_{\pm1.57}$ | $33.33_{\pm1.62}$ | 45.56 |
| HybridLLM (Ding et al., 2024) | L3B&G4.1 | $52.9_{\pm0.94}$ | $43_{\pm0.82}$ | $22.22_{\pm1.57}$ | $36.67_{\pm0.62}$ | 38.70 |
| DoT (Shao et al., 2025) | L3B&G4.1 | $50.54_{\pm3.04}$ | $66_{\pm1.63}$ | $21.11_{\pm3.14}$ | $48.33_{\pm1.89}$ | 46.50 |
| HybridFlow (Ours) | L3B&G4.1 | $53.33_{\pm2.03}$ | $\underline{72.54_{\pm0.65}}$ | $36.67_{\pm1.57}$ | $\underline{58.83_{\pm1.48}}$ | $\underline{55.34}$ |

*AIME24*; *MMLU-Pro* (Wang et al., 2024), and *LiveBench-Reasoning* (White et al., 2025). We select three metrics that jointly capture reasoning quality and system efficiency. (i) *Acc* measures the correctness of reasoning by comparing model outputs with gold answers across all benchmarks. (ii) $C_{\text{time}}$ denotes the end-to-end inference latency per query, including decomposition, routing, and execution. (iii) $C_{\text{API}}$ quantifies the API cost consumed by cloud LLM calls, reflecting both API efficiency and cost.

**Baselines.** We compare HybridFlow with representative task decomposition methods from single-model and edge-cloud collaborative paradigms, as well as a direct prompting baseline. For single-model methods, we select *CoT* (Wei et al., 2022), *SoT* (Ning et al., 2024) and *PASTA* (Jin et al., 2025). For collaborative inference, we select *HybridLLM* (Ding et al., 2024), and *DoT* (Shao et al., 2025).

**Implementation Details.** HybridFlow deploys `Llama3.2-3B` on the edge device in two roles: (i) as the *planner* $M_P$ that decomposes each query into a task-level DAG, and (ii) as the edge executor $M_{\text{edge}}$ that runs subtasks not offloaded. Each subtask is encoded with `qwen3-embedding-0.6b` into an embedding $z_i$, which is fed to a lightweight router: a two-hidden-layer MLP that predicts an offline utility estimate $\hat{u}_i$. The router is offline warm-started with AdamW (learning rate $1 \times 10^{-4}$) by regressing to profiled utility targets. During inference, HybridFlow performs online adaptation with dual-threshold updates that track budget consumption; an online calibration head is updated from partial feedback via a contextual-bandit strategy. Routing decisions use the current threshold to assign each ready subtask to either $M_{\text{edge}}$ or the cloud model $M_{\text{cloud}}$ (`GPT-4.1`, via API). All edge-side computation (planning, local execution, and embedding) runs on a single NVIDIA RTX 3090 GPU. All LLMs use a fixed temperature of $0.6$.

## 4.2. Results

**Task decomposition enhances multi-step reasoning performance.** Across all benchmarks, methods that explicitly decompose tasks into structured intermediate steps show clear advantages over direct prompting. As shown in Table 1, CoT with GPT-4.1 attains the highest overall accuracy among non-Prompt methods (**58.99**%), confirming the strong benefits of stepwise reasoning. HybridFlow closely follows with an average accuracy of 55.34%, outperforming SoT and PASTA variants while remaining competitive with the strongest single-model reasoning approach. On GPQA and MMLU-Pro, HybridFlow achieves 53.33% and 72.54%, respectively, narrowing the accuracy gap with all-cloud CoT to 4–7% while using less than half the API cost. This demonstrates that HybridFlow's planner produces decomposition structures not only logically aligned with the tasks but also highly executable by collaborating edge and cloud models.

**Parallel execution substantially reduces end-to-end latency.** HybridFlow's dependency-aware DAG planning enables fine-grained concurrency during subtask execution. This design significantly reduces wall-clock latency compared to sequential or coarse-grained hybrid pipelines. As reported in Table 2, HybridFlow achieves an average $C_{\text{time}}$ of 17.48 s, outperforming HybridLLM (24.45 s) by 28.5% and even improving over the sequentially constrained DoT baseline (18.32 s). The latency reduction stems from parallel execution of independent subtasks: on MMLU-Pro, where the planner exposes the most parallelism, Hybrid-Flow completes inference in 11.85 s compared with DoT's 11.00 s and HybridLLM's 14.90 s, despite using comparable or fewer cloud resources. These results confirm that exploiting parallelism within logically valid execution windows can effectively offset the overhead of multi-step reasoning, leading to faster and more responsive inference.

*Table 2.* Efficiency comparison of HybridFlow and baselines on four reasoning benchmarks. Lower values indicate better efficiency for both end-to-end inference time ($C_{\text{time}}$, in seconds) and cloud API cost ($C_{\text{API}}$). Bold denotes the best and underline marks the second-best performance among edge-cloud collaboration baselines. Direct Prompt rows (shaded) serve as reference points and are excluded from ranking. HybridFlow consistently improves latency and cloud usage through dependency-aware parallel execution and adaptive routing.

| Method | Model | Metric | GPQA | MMLU-Pro | AIME24 | LiveBench-Reasoning | Avg. (↓) |
|---|---|---|---|---|---|---|---|
| Direct Prompt | L3B | $C_{\text{time}}$ | 6.61±0.50 | 7.03±0.64 | 9.92±1.51 | 13.34±0.40 | 9.23 |
| Direct Prompt | G4.1 | $C_{\text{time}}$ | 15.26±1.85 | 11.77±0.18 | 50.44±1.64 | 36.77±1.61 | 28.56 |
| Direct Prompt | L3B | $C_{\text{API}}$ | – | – | – | – | – |
| Direct Prompt | G4.1 | $C_{\text{API}}$ | 0.0094 | 0.0060 | 0.0256 | 0.0181 | 0.0148 |
| CoT (Wei et al., 2022) | L3B | $C_{\text{time}}$ | 11.99±0.25 | 10.87±0.45 | 22.76±4.78 | 14.00±0.17 | 14.91 |
| CoT | L3B | $C_{\text{API}}$ | – | – | – | – | – |
| CoT | G4.1 | $C_{\text{time}}$ | 18.26±2.49 | 19.35±0.22 | 56.70±2.66 | 29.77±0.79 | 31.02 |
| CoT | G4.1 | $C_{\text{API}}$ | 0.0185 | 0.0115 | 0.0445 | 0.0330 | 0.0269 |
| SoT (Ning et al., 2024) | L3B | $C_{\text{time}}$ | 18.55±0.31 | 10.95±0.48 | 15.20±0.85 | 14.61±0.78 | 14.83 |
| SoT | L3B | $C_{\text{API}}$ | – | – | – | – | – |
| SoT | G4.1 | $C_{\text{time}}$ | 16.27±1.57 | 11.43±0.03 | 29.52±0.56 | 20.87±0.81 | 19.52 |
| SoT | G4.1 | $C_{\text{API}}$ | 0.0154 | 0.0095 | 0.0328 | 0.0206 | 0.0196 |
| PASTA (Jin et al., 2025) | L3B | $C_{\text{time}}$ | 8.77±1.19 | 14.15±0.68 | 12.43±1.24 | 15.65±0.58 | 12.75 |
| PASTA | L3B | $C_{\text{API}}$ | – | – | – | – | – |
| PASTA | G4.1 | $C_{\text{time}}$ | 12.21±1.72 | 8.76±0.76 | 21.37±1.65 | 19.14±1.29 | 15.37 |
| PASTA | G4.1 | $C_{\text{API}}$ | 0.0262 | 0.0179 | 0.0474 | 0.0338 | 0.0313 |
| HybridLLM (Ding et al., 2024) | L3B&G4.1 | $C_{\text{time}}$ | 15.96±1.74 | 14.90±0.40 | 40.11±2.25 | 26.82±1.65 | 24.45 |
| HybridLLM | L3B&G4.1 | $C_{\text{API}}$ | 0.0160 | **0.0050** | 0.0168 | 0.0135 | 0.0128 |
| DoT (Shao et al., 2025) | L3B&G4.1 | $C_{\text{time}}$ | 15.79±0.67 | 11.00±0.45 | 29.91±2.50 | 16.59±0.80 | 18.32 |
| DoT | L3B&G4.1 | $C_{\text{API}}$ | 0.0078 | 0.0056 | 0.0138 | **0.0087** | 0.009 |
| HybridFlow (Ours) | L3B&G4.1 | $C_{\text{time}}$ | 15.24±0.30 | 11.85±0.38 | 26.40±1.54 | 16.41±0.59 | **17.48** |
| HybridFlow (Ours) | L3B&G4.1 | $C_{\text{API}}$ | **0.0075** | 0.0052 | **0.0135** | 0.0091 | **0.0088** |

**HybridFlow delivers the best accuracy-efficiency trade-off.** While HybridFlow's accuracy approaches the top-performing CoT with GPT-4.1, it does so with dramatically lower API consumption and latency. Table 2 shows that HybridFlow achieves the lowest average $C_{\text{API}}$ (**0.0088**) among all collaborative baselines, indicating high API efficiency in cloud usage. Relative to DoT, HybridFlow reduces average API cost by 2.2% (0.009→0.0088) and latency by 4.6% while improving accuracy by 8.8% (46.50%→55.34%); relative to HybridLLM, it improves accuracy by 16.6% (38.70%→55.34%) while cutting latency by 28.5% and API cost by 31.3%. When considering both accuracy and efficiency jointly, HybridFlow consistently dominates alternative hybrid systems: it improves accuracy relative to HybridLLM and DoT while simultaneously reducing latency and API cost. This demonstrates that HybridFlow effectively balances the strengths of edge and cloud models through adaptive routing and parallelized task execution, achieving a superior overall accuracy-efficiency frontier.

### 4.3. Ablation Study

In this section, we conduct a series of ablation studies to validate the effectiveness of our adaptive allocation mechanism. Our goal is to demonstrate that the router is crucial for

achieving a balance between latency, cost, and performance.

**Router for Subtask Allocation.** Table 3 compares our adaptive router with naive and deterministic allocation strategies. Executing all subtasks on the edge model (Edge) eliminates cloud cost but attains only 25.54% accuracy, while full cloud execution (Cloud) achieves the highest accuracy (57.28%) at higher latency (18.26 s) and API cost ($C_{\text{API}}$=0.0185). The Random baseline reduces API cost ($C_{\text{API}}$=0.0075) but yields lower accuracy (46.00%), indicating inefficient use of cloud budget: without utility-guided selection, it squanders API calls on low-impact subtasks while neglecting high-value ones. A fixed-threshold policy improves accuracy to 51.62% with modest cost, yet remains inferior to our learned routing. HybridFlow-Chain further shows that routing alone is insufficient: disabling dependency-aware parallelism reduces performance (50.62% accuracy; utility 0.6095). In contrast, HybridFlow achieves 53.33% accuracy at the same low API cost as Random ($C_{\text{API}}$=0.0075) and attains the highest utility (0.7940), demonstrating a better accuracy-cost trade-off under our unified metric.

**Offload Ratio between Edge and Cloud.** To characterize the router's within-query allocation, we count how many

*Table 3.* Ablation of routing strategies on GPQA. We report the subtask offload rate, end-to-end accuracy, latency, cloud API cost $C_{API}$, normalized total cost $c$ (lower is better), and the unified utility $u$ (higher is better). HybridFlow-Chain disables DAG parallelism and executes subtasks sequentially, isolating the effect of routing from scheduling. Our HybridFlow achieves the best accuracy-cost trade-off, yielding the highest utility.

| Method | Offload Rate (%) | Accuracy (%) | Latency (s) | API Cost ($) | Norm. Cost $c$ ($\downarrow$) | Utility $u$ ($\uparrow$) |
|---|---|---|---|---|---|---|
| Edge (*Llama3.2-3B*) | 0 | 25.54 | 11.99 | 0 | – | – |
| Cloud (*GPT-4.1*) | 100 | **57.28** | 18.26 | 0.0185 | 0.7760 | 0.4090 |
| Random (*Llama3.2-3B + GPT-4.1*) | 42.1 | 46.00 | **15.15** | 0.0075 | **0.3455** | 0.5922 |
| Fixed Threshold ($\tau_0 = 0.5$) | 41.18 | 51.62 | 15.88 | 0.0088 | 0.4145 | 0.6292 |
| HybridFlow-Chain | 40.81 | 50.62 | 16.12 | 0.0082 | 0.4115 | 0.6095 |
| HybridFlow-NoCalibration | 40.85 | 51.85 | 15.21 | 0.0079 | 0.3585 | 0.4558 |
| HybridFlow (Ours) | 40.48 | 53.33 | 15.24 | **0.0075** | 0.3500 | **0.7940** |

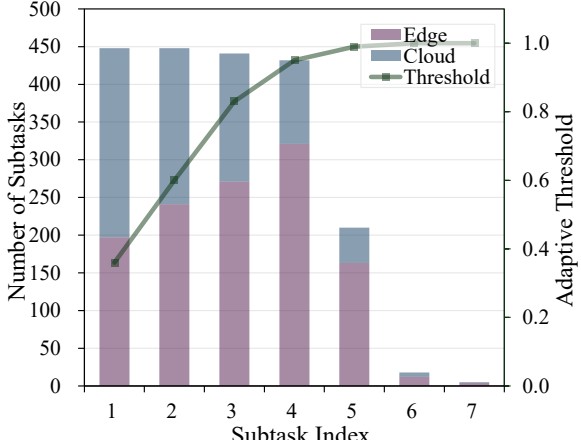

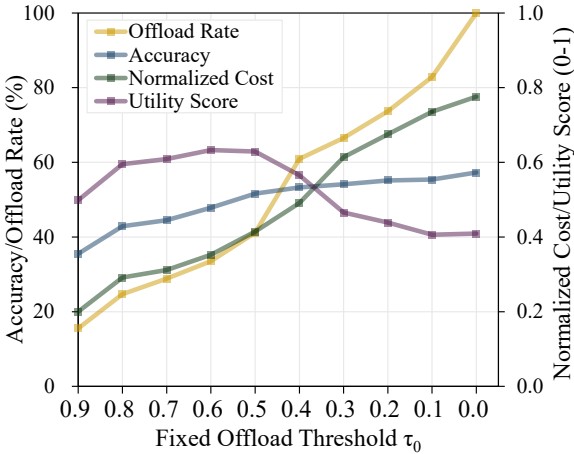

*Figure 3.* Distribution of executed subtasks between the edge and cloud models across subtask positions on GPQA. Bars show the number of subtasks executed on the edge (purple) and on the cloud (blue) at each subtask index, and the line shows the average adaptive threshold at that position.

*Figure 4.* Performance–cost trends under different fixed offload thresholds $\tau_0$ on the GPQA benchmark. Increasing $\tau_0$ makes the router more conservative, leading to lower offload rate and cost but gradually reducing accuracy.

subtasks are executed on the edge vs. the cloud at each subtask position. Figure 3 shows a clear position-dependent pattern rather than uniform offloading. At early positions, the adaptive threshold is relatively low and the remaining budget is large, making cloud execution more likely. As reasoning proceeds, the threshold increases and eventually saturates; correspondingly, routing shifts toward the edge, with few cloud calls at later positions. Meanwhile, the total number of subtasks decreases at deeper positions, suggesting that many queries resolve key reasoning steps early. Overall, HybridFlow concentrates cloud usage on early, high-impact subtasks and relies on the edge model for downstream subtasks as the routing criterion becomes stricter. This temporal allocation pattern naturally emerges from the budget-aware design: the router front-loads expensive cloud calls to subtasks that most benefit from stronger reasoning, then progressively shifts to edge execution as remaining budget tightens, ensuring that limited cloud resources are spent where they yield the greatest returns.

**Effect of Fixed Offload Thresholds.** To isolate the role of the base threshold, we vary the fixed offload threshold $\tau_0$ in Eq. 11 without dynamic resource adaptation and measure accuracy, offload rate, latency, and cost. As shown in Figure 4 and Table 9, increasing $\tau_0$ makes routing more conservative: the offload rate decreases monotonically from 100.00% ($\tau_0=0$) to 0.00% ($\tau_0=1$), and the normalized cost decreases accordingly from $c=0.7760$ to $c=0.2000$ at $\tau_0=0.9$ (and is undefined when no cloud calls are made). Accuracy decreases overall across the same range, from 57.28% at $\tau_0=0$ to 25.54% at $\tau_0=1$. This consistent trade-off confirms that the predicted utilities $\hat{u}_i$ and the normalized costs $c_i$ are well calibrated on a common $[0,1]$ scale, yielding meaningful decision boundaries.

Notably, the utility score peaks at $\tau_0=0.6$ with $u=0.6329$, where the router maintains 47.85% accuracy at a moderate normalized cost ($c=0.3525$) and an offload rate of 33.51%. Moving to a looser threshold (e.g., $\tau_0=0.5$) increases both accuracy (51.62%) and cost ($c=0.4145$) but slightly reduces

*Table 4.* GPQA results under a swapped edge/cloud model pair (Qwen2.5-7B on edge; DeepSeek-V3 on cloud).

| Method | Accuracy | API Cost ($\times 10^{-3}$\$) | Latency (s) |
|---|---|---|---|
| All-Edge CoT (Qwen2.5-7B) | 34% | NA | 19.52 |
| All-Cloud CoT (DeepSeek-V3) | 59% | 6.70 | 61.00 |
| HybridLLM (Ding et al., 2024) | 47% | 3.63 | 47.87 |
| DoT (Shao et al., 2025) | 49% | 1.80 | 40.90 |
| HybridFlow (Ours) | **53%** | **1.16** | **36.86** |

utility (0.6292), while more aggressive offloading further raises cost and eventually decreases utility (e.g., $u$=0.5652 at $\tau_0$=0.4 and $u$=0.4090 at $\tau_0$=0). These results indicate that a single global threshold can perform well only within narrowly constrained operating conditions: even a 0.1 shift in $\tau_0$ can alter the offload rate by 8–18% and change utility by up to 6%, making fixed thresholds brittle in practice.

In contrast, HybridFlow achieves a substantially higher utility (0.7940 in Table 3) by dynamically adapting routing decisions to the evolving budget, outperforming any fixed-threshold setting while avoiding both excessive offloading and premature underutilization.

**Generality across model pairs.** To test whether Hybrid-Flow depends on a particular edge–cloud instantiation, we conduct a *model-pair swap* on GPQA: we replace the edge model with **Qwen2.5-7B** and the cloud model with **DeepSeek-V3**, while keeping the rest of the system *unchanged* (task decomposition, routing, scheduling, and evaluation protocol). All baselines are evaluated under the same swapped pair for a fair comparison. As shown in Table 4, HybridFlow maintains a strong cost–latency–accuracy trade-off: compared to DoT, it improves accuracy by 4 points (49%→53%) while reducing API cost by 35.6% and latency by 9.9%; compared to HybridLLM, it improves accuracy by 6 points and reduces cost and latency by 68.0% and 23.0%, respectively. This suggests HybridFlow's subtask-level routing and dependency-aware scheduling are not specialized to the Llama3.2-3B/GPT-4.1 pair and can transfer to different model scales with minimal changes. The consistent improvement pattern across both model pairs indicates that the gains stem from the framework's structural design—fine-grained decomposition and adaptive allocation—rather than from idiosyncratic interactions between specific models.

## 5. Conclusion

In this paper, we introduce HybridFlow, a resource-adaptive inference framework that formulates fast, token-efficient collaborative reasoning as a sequential decision process. By decomposing complex queries into a dependency-aware DAG, HybridFlow optimizes the reasoning path and facilitates parallel subtask execution. Our resource-aware subtask router, which moves beyond rigid, coarse-grained alloca-

tion, enables HybridFlow to adaptively assign subtasks to edge or cloud resources. This process allows the framework to achieve superior performance while balancing inference time, token usage, and real-time budget states. On comprehensive evaluations including GPQA, MMLU-Pro, AIME24, and LiveBench-Reasoning, HybridFlow effectively outperforms sequential and coarse-grained baselines, demonstrating significant reductions in both end-to-end latency and overall token consumption. These results demonstrate the promise of our adaptive, parallel-aware routing framework for orchestrating efficient edge-cloud LLM.

## Acknowledgements

This research was supported by: 1) Project P0061995 under the Financial Support for Non-PAIR Research Centres, funded by the Hong Kong PolyU (UGC); 2) Project P0049179 under the Innovation and Technology Fund - Guangdong-Hong Kong Technology Cooperation Funding Scheme (ITF-TCFS), funded by the Innovation and Technology Commission (Funding Body Ref. No. GHP/386/23SZ).

## Impact Statement

This paper presents HybridFlow, a framework for efficient edge–cloud collaborative inference. We expect several positive societal impacts. By reducing reliance on cloud-only inference and enabling more computation at the edge, our work can lower energy consumption and API costs, making LLM-powered applications more accessible to resource-constrained users and regions with limited connectivity. Keeping sensitive subtasks on-device also offers privacy benefits by reducing the volume of data transmitted to cloud providers. Moreover, by cutting redundant data transfers between edge devices and remote data centers, we can reduce the energy cost and carbon footprint.

At the same time, we recognize potential risks. More efficient inference could accelerate the deployment of LLMs in high-stakes domains before adequate safeguards are in place, or exacerbate environmental impacts if demand grows to offset efficiency gains (the Jevons paradox). We encourage practitioners to evaluate HybridFlow's behavior on their specific tasks before deployment and to maintain human oversight for consequential decisions.

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

# A. Notations

Table 5. Notations and definitions used in HybridFlow.

| Notation | Definition |
|---|---|
| $Q$ | User-issued query. |
| $M_{\text{edge}}$ | Small edge model used for local inference. |
| $M_{\text{cloud}}$ | Large cloud LLM accessed via API. |
| $M_P$ | Edge-deployed planner model (Llama3.2–3B). |
| $T = \{t_i\}_{i=1}^n$ | Set of subtasks produced from query $Q$. |
| $E$ | Directed edge set encoding prerequisite relations between subtasks. |
| $G(Q) = (T, E)$ | Task-level decomposition DAG for query $Q$. |
| $n$ | Number of subtasks in the decomposition. |
| $\Delta q_i$ | Expected accuracy gain of executing $t_i$ on $M_{\text{cloud}}$ vs. $M_{\text{edge}}$. |
| $\Delta l_i$ | Additional latency cost (seconds) of executing $t_i$ on $M_{\text{cloud}}$. |
| $\Delta k_i$ | Additional API usage cost (tokens or price units) of executing $t_i$ on $M_{\text{cloud}}$. |
| $l_{\text{max}}^{\text{sub}}$ | Per-subtask upper bound on additional latency for normalization. |
| $k_{\text{max}}^{\text{sub}}$ | Per-subtask upper bound on additional API cost for normalization. |
| $c_i$ | Normalized cost of offloading $t_i$ [Eq. (1)]. |
| $r_i$ | Routing decision for $t_i$: 1 if offloaded to $M_{\text{cloud}}$, 0 otherwise. |
| $\mathbf{r} = (r_1, \ldots, r_n)$ | Routing vector for all subtasks of a query. |
| $C_{\text{used}}(t) = \sum_{j \leq t} r_j c_j$ | Cumulative normalized cost at time $t$. |
| $C_{\text{max}}$ | Total normalized resource budget for a query [Eq. (3)]. |
| $u_i$ | Utility (normalized benefit–cost ratio) of offloading $t_i$ [Def. 3.2]. |
| $\varepsilon$ | Small positive constant (e.g., $10^{-4}$) for numerical stability in $u_i$. |
| $\tau_0$ | Base routing threshold. |
| $\tau_t$ | Adaptive routing threshold at time $t$ [Eq. (11)]. |
| $z_i$ | Semantic embedding of subtask $t_i$. |
| $f_\theta$ | Router network parameterized by $\theta$. |
| $\hat{u}_i$ | Predicted utility of offloading $t_i$. |
| $\sigma(\cdot)$ | Sigmoid activation function used in the router. |
| $\mathcal{L}(\theta)$ | Training loss for the router parameters $\theta$. |
| $N$ | Number of profiled subtasks used for router training. |
| $\mathcal{Q}$ | Scheduler queue of *ready* subtasks in the DAG. |

# B. Optimization View of HybridFlow

This section provides a rigorous optimization formulation of the HybridFlow routing problem and establishes the theoretical underpinnings of the utility-based router introduced in Sec. 3.3. We formalize the allocation problem as a 0–1 knapsack problem, derive its Lagrangian relaxation, and show how the adaptive threshold mechanism used in HybridFlow naturally emerges as a primal–dual update for this relaxation.

## B.1. 0–1 Knapsack Formulation

For a query decomposed into subtasks $T = \{t_1, \ldots, t_n\}$, recall that $\Delta q_i$ is the accuracy gain from offloading $t_i$ to the cloud, $c_i \in [0, 1]$ is the normalized resource cost, and $r_i \in \{0, 1\}$ indicates whether $t_i$ is offloaded. Let $C_{\text{max}} \in [0, 1]$ denote the normalized per-query resource budget.

The routing problem in Sec. 3.3 can be written as:

$$
\begin{aligned}
\max_{\mathbf{r} \in \{0,1\}^n} \quad & \sum_{i=1}^n r_i \Delta q_i \\
\text{s.t.} \quad & \sum_{i=1}^n r_i c_i \leq C_{\text{max}},
\end{aligned}
\tag{15}
$$

which is exactly the *0–1 knapsack problem*, with each subtask $t_i$ corresponding to an item of value $\Delta q_i$ and weight $c_i$. This formulation provides both: (i) a principled objective for allocation, and (ii) an optimal oracle via dynamic programming for evaluation.

## B.2. Lagrangian Relaxation

We obtain a continuous relaxation of the knapsack by dualizing the budget constraint. Introducing a Lagrange multiplier $\lambda \geq 0$, the Lagrangian becomes:

$$\mathcal{L}(\mathbf{r}, \lambda) = \sum_{i=1}^{n} r_i \Delta q_i - \lambda \Big( \sum_{i=1}^{n} r_i c_i - C_{\max} \Big). \tag{16}$$

Expanding the expression yields:

$$\mathcal{L}(\mathbf{r}, \lambda) = \lambda C_{\max} + \sum_{i=1}^{n} r_i \big( \Delta q_i - \lambda c_i \big). \tag{17}$$

For a fixed $\lambda$, the optimization decomposes across subtasks:

$$r_i^\star(\lambda) = \arg \max_{r_i \in \{0,1\}} r_i (\Delta q_i - \lambda c_i) = \mathbb{I}[\Delta q_i - \lambda c_i > 0], \tag{18}$$

where $\mathbb{I}[\cdot]$ is the indicator function.

Thus the relaxed problem prescribes the following thresholding rule:

$$\text{Offload } t_i \quad \Longleftrightarrow \quad \frac{\Delta q_i}{c_i} > \lambda. \tag{19}$$

Here $\lambda$ plays the role of a *shadow price* of resource consumption: subtasks with benefit–cost ratio above $\lambda$ should be offloaded.

## B.3. Primal–Dual Dynamics and Adaptive Thresholding

HybridFlow performs *online* allocation as subtasks become ready. A natural approach is to maintain a time-varying estimate of the shadow price $\lambda_t$ and update it based on cumulative consumption. A standard primal–dual update for the constraint $\sum_i r_i c_i \leq C_{\max}$ is:

$$\lambda_{t+1} = \big[ \lambda_t + \eta (C_{\text{used}} - C_{\max}) \big]_+, \tag{20}$$

where $\eta > 0$ is a step size and $[\cdot]_+$ denotes projection onto $[0, \infty)$.

HybridFlow's adaptive threshold in Eq. (11) of the main text,

$$\tau_t = \text{clip}\Big( \tau_0 + \frac{k_{\text{used}}}{2K_{\max}^{\text{global}}} + \frac{l_{\text{used}}}{2L_{\max}^{\text{global}}}, 0, 1 \Big), \tag{21}$$

is precisely an instance of a primal–dual update:

- the additive terms track *dual pressure* from API cost and latency,

- the clipping corresponds to projected dual ascent,

- the threshold $\tau_t$ plays the role of the shadow price $\lambda_t$ in Eq. (19).

Thus HybridFlow's routing rule,

$$\hat{u}_i > \tau_t \quad \Longleftrightarrow \quad \frac{\Delta q_i}{c_i} \gtrsim \lambda_t, \tag{22}$$

is an online approximation to the Lagrangian decision rule in Eq. (18).

### B.4. Learned Approximation to the Optimal Policy

HybridFlow does not observe $\Delta q_i$ or $c_i$ at inference time. Instead it uses learned utility estimates $\hat{u}_i \approx u_i$ obtained from a lightweight MLP. Under mild smoothness assumptions on the embedding mapping and the true utility function, the resulting allocation rule

$$r_i = \mathbb{I}[\hat{u}_i > \tau_t] \tag{23}$$

approximates the relaxed knapsack-optimal policy while ensuring online compliance with global resource budgets. The adaptive threshold thereby provides principled control over budget usage without requiring explicit dynamic programming.

### B.5. Implications

This optimization analysis yields several insights:

- **Interpretability:** Each routing decision reduces to comparing a predicted marginal utility with a time-varying shadow price.

- **Optimality Structure:** The DP oracle defines an upper bound for achievable allocation quality; HybridFlow's router approximates this solution in a computationally lightweight manner.

- **Budget Compliance:** The adaptive threshold implements a projected dual ascent rule, increasing conservativeness as resource usage grows.

- **Scalability:** Because the relaxed problem is decomposable, HybridFlow can make routing decisions independently across subtasks while preserving global budget coherence.

This provides a principled foundation for the design of HybridFlow's routing mechanism and clarifies its connection to classical combinatorial optimization.

## C. Implementation Details

**Subtasks and decomposition DAG.** Given an input query $Q$, HybridFlow represents a reasoning plan as a directed acyclic graph (DAG) $G(Q) = (T, E)$, where $T = \{t_1, \ldots, t_n\}$ is the set of subtasks and $E \subseteq T \times T$ encodes prerequisite relations.

**Definition C.1** (Subtask). A subtask is a tuple

$$t_i = (d_i, P_i, \tau_i),$$

where (i) $d_i$ is a natural-language description of the operation to be performed (e.g., "Check whether the inverse property holds"), (ii) $P_i \subseteq \{1, \ldots, n\} \setminus \{i\}$ is the index set of its prerequisite subtasks, and (iii) $\tau_i \in \{\text{EXPLAIN}, \text{ANALYZE}, \text{GENERATE}\}$ is a role label that follows the Explain–Analyze–Generate (EAG) metaprompt structure.

For convenience, we write $t_j \rightarrow t_i$ whenever $j \in P_i$. The edge set is then $E = \{(t_j, t_i) : j \in P_i, i = 1, \ldots, n\}$.

**Definition C.2** (Valid decomposition). A decomposition of $Q$ is a DAG $G(Q) = (T, E)$ with $T = \{t_i\}_{i=1}^{n}$ that satisfies:

1. (Acyclicity) $G(Q)$ is acyclic.

2. (Rooted plan) There exists a unique root node $t_{\text{root}}$ with $P_{\text{root}} = \emptyset$ and $\tau_{\text{root}} = \text{EXPLAIN}$.

3. (Reachability) Every subtask is reachable from the root: for all $i$, there exists a directed path $t_{\text{root}} \rightsquigarrow t_i$.

4. (Well-formed outputs) At least one node is labeled GENERATE, and every GENERATE node has no outgoing edges (i.e., it is a sink in $G(Q)$). We require exactly one GENERATE sink node that produces the final answer.

5. (Size constraint) The number of subtasks is bounded by a constant $n \leq n_{\max}$ (we use $n_{\max} = 7$ in experiments), which controls planner overhead.

6. (Dependency consistency) For any edge $t_j \rightarrow t_i$, the output of $t_j$ is referenced by $t_i$. Concretely, each subtask $t_i$ declares a set of required symbols $\text{Req}(t_i)$ and produced symbols $\text{Prod}(t_i)$ (parsed from the XML plan); we require $\text{Req}(t_i) \subseteq \bigcup_{t_j \in P_i} \text{Prod}(t_j)$.

We denote by $\mathcal{G}(Q)$ the set of all valid decompositions for query $Q$.

The planner $M_P$ induces a (stochastic) mapping

$$Decompose : Q \mapsto G(Q) \in \mathcal{G}(Q),$$

implemented as a prompt-based generation of an XML-formatted plan followed by a deterministic parsing and validation procedure.

**Validation and repair.** After parsing the XML plan, we validate it using the rules in Definition C.2 (acyclicity, rootedness, reachability, well-formed outputs, size constraint) and an additional dependency-consistency check based on $\text{Req}(\cdot)/\text{Prod}(\cdot)$ symbols. If validation fails, we apply a bounded, deterministic repair procedure: (i) remove ill-typed edges (violating dependency consistency), (ii) break cycles by removing the lowest-confidence edge[1], (iii) enforce rootedness/reachability by attaching orphan nodes to the root, (iv) if the plan is still invalid after $R_{\max}$ iterations, we fall back to a chain plan (sequential execution). We use $R_{\max} = 2$ in all experiments to bound overhead.

*Table 6.* Planner DAG validity and repair statistics across benchmarks. VALID denotes plans passing validation without repair; REPAIRED denotes plans fixed within $R_{\max}$ iterations; FALLBACK denotes sequential chain plans.

| Benchmark | Valid (%) | Repaired (%) | Fallback (%) | #nodes (avg) |
|---|---|---|---|---|
| GPQA | 76 | 14 | 10 | 4.47 |
| LiveBench-Reasoning | 78 | 13 | 9 | 4.32 |

Table 6 reports the planner's decomposition reliability across benchmarks. We observe that the majority of queries yield a valid DAG without any repair (76–78%), indicating that the prompt-based planner typically produces well-formed, executable dependency structures under our constraints (Definition C.2). For an additional 13–14% of queries, the initial plan violates one or more validity checks (e.g., minor reachability or acyclicity issues) but can be deterministically repaired within $R_{\max}$ iterations; these repaired plans are subsequently executed as DAGs. Only 9–10% of queries trigger the fallback-to-chain mechanism after bounded repair attempts, which guarantees robustness while keeping the planner overhead controlled. Finally, among the executed DAG plans (VALID+REPAIRED), the average number of subtasks is approximately 4.3–4.5, suggesting that the planner produces moderately granular decompositions that expose parallelism without excessive fragmentation, consistent with our $n_{\max} = 7$ cap.

To analyze how fallback cases affect the overall accuracy and latency numbers, we conduct additional test on full GPQA in Table 7. We find comparable accuracy in fallback cases, but a modest latency gap due to their chain execution structure. Importantly, fallback cases account for only a small fraction of the total samples, so their impact on the overall DAG-based evaluation is limited.

*Table 7.* Fallback case analysis on GPQA benchmark.

| GPQA | Number | Accuracy(%) | Latency(s) | API cost($) |
|---|---|---|---|---|
| valid | 341 | 53.5 | 15.27 | 0.00745 |
| repaired | 63 | 52.4 | 15.33 | 0.00767 |
| fallback | 44 | 52.8 | 16.73 | 0.00762 |

**Quality and Cost Estimation.** To train the proposed resource-aware router, we construct an offline profiling dataset from 2,000 sampled queries drawn from two benchmarks: MMLU-Pro (different from the main test samples) and Math500 (covering general knowledge reasoning, targeting structured, multi-step reasoning). For each query, we perform the complete pipeline of task decomposition, subtask allocation, and execution using both the edge and cloud models. During this process, we record the response quality, inference latency, and API cost for each subtask.

For each subtask $i$, we construct paired executions that differ only in whether subtask $i$ is processed on the cloud or on the edge while keeping the rest of the pipeline unchanged. We then evaluate the two resulting end-to-end answers with

---

[1]We define confidence as the planner's self-reported score per edge in the XML; when unavailable, we use a fixed priority order.

a task-specific verifier (e.g., exact-match / multiple-choice correctness or a rubric-based judge) to compute the marginal correctness gain of cloud execution. Concretely, $\Delta q_i$ is defined as the difference between the final outcome scores of the two trajectories (cloud vs. edge for subtask $i$). This offline profiling is performed once on a held-out set and incurs no additional overhead during online inference.

In practice, each query is decomposed into 4–5 subtasks on average (we constrain the planner to generate fewer than 7 subtasks). For each query, we execute *every* subtask once on the edge model and once on the cloud model, and cache the resulting subtask outputs. Since the planner is fixed and does not re-plan conditioned on upstream outputs, we can efficiently construct mixed edge–cloud executions by recombining these cached outputs. We then sample a set of routing vectors and evaluate the final task outcome for each recombined execution. Finally, we estimate the contribution of each subtask by averaging its marginal effect over the sampled contexts, i.e., comparing outcomes when toggling that subtask between edge and cloud while keeping the remaining routing choices unchanged. Overall, this reuse-and-recombine strategy provides a reasonable approximation of subtask-level effects while substantially reducing profiling cost compared to rerunning the full pipeline for every counterfactual.

Let $\Delta q_i$ denote the expected quality improvement when offloaded to the cloud, and $(\Delta l_i, \Delta k_i)$ represent the additional latency and API cost, respectively. We define the normalized cost term as:

$$c_i = \frac{1}{2} \cdot \frac{\Delta l_i}{10} + \frac{1}{2} \cdot \frac{\Delta k_i}{0.02}, \tag{24}$$

where the constants 10 and 0.02 correspond to the normalization scales of latency (seconds) and API cost ($), ensuring $c_i \in [0, 1]$ across all profiled subtasks.

We evaluate the overhead of profiling here. Accordingly, our held-out audit should be interpreted as evidence of a one-time offline cost, rather than online runtime overhead. We report the corresponding profiling latency and API cost in Table 8. Since the router itself is lightweight, we believe this profiling cost remains affordable in practice.

Table 8. Profiling latency and API cost on the GPQA benchmark.

| Profile audit | Value |
|---|---|
| queries / subtasks | 2000 / 12400 |
| decompose total / avg | 3240.2s / 1.62s |
| paired execution total / avg | 28595.4s / 14.30s |
| total offline / avg | 31835.6s / 15.92s |
| LLM API cost/ avg | $1.36 / $6.8e-4 |
| calls per query | 6.2 |

**Training Objective.**   Each subtask is annotated with its measured $(\Delta q_i, \Delta l_i, \Delta k_i)$ values, and the corresponding target utility score is computed as:

$$u_i = \text{clip}\left( \frac{\Delta q_i}{c_i + \varepsilon}, \, 0, \, 1 \right), \tag{25}$$

where $\varepsilon$ prevents division by zero and $u_i$ reflects the normalized benefit–cost ratio. The router model $f_\theta$ is trained to regress this target utility from the subtask embedding $z_i$ via a mean-squared loss:

$$\mathcal{L}(\theta) = \frac{1}{N} \sum_{i=1}^{N} \left( \hat{u}_i - u_i \right)^2, \tag{26}$$

This offline training enables the router to approximate the benefit–cost trade-off without requiring online supervision during inference.

**Adaptive Threshold Configuration.**   At inference time, the router compares its predicted utility $\hat{u}_i$ against an adaptive threshold $\tau_t$ to decide whether to offload a subtask. The threshold evolves with real-time resource usage as:

$$\tau_t = \text{clip}\left( \tau_0 + \frac{k_{\text{used}}}{2K_{\text{max}}} + \frac{l_{\text{used}}}{2L_{\text{max}}}, \, 0, \, 1 \right), \tag{27}$$

*Table 9.* Performance–cost trends under different fixed offload thresholds $\tau_0$ on the GPQA benchmark.

| Threshold | Offload Rate (%) | Accuracy (%) | Latency (s) | API Cost ($) | Normalized Cost ($\downarrow$) | Utility ($\uparrow$) |
|---|---|---|---|---|---|---|
| 1 | 0.00 | 25.54 | 11.99 | 0 | N/A | N/A |
| 0.9 | 15.51 | 35.51 | 13.89 | 0.0042 | 0.2000 | 0.4985 |
| 0.8 | 24.67 | 42.89 | 14.87 | 0.0059 | 0.2915 | 0.5952 |
| 0.7 | 28.85 | 44.51 | 15.02 | 0.0064 | 0.3115 | 0.6090 |
| 0.6 | 33.51 | 47.85 | 15.39 | 0.0073 | 0.3525 | 0.6329 |
| 0.5 | 41.18 | 51.62 | 15.88 | 0.0088 | 0.4145 | 0.6292 |
| 0.4 | 60.95 | 53.29 | 16.56 | 0.0105 | 0.4910 | 0.5652 |
| 0.3 | 66.51 | 54.13 | 17.87 | 0.0128 | 0.6140 | 0.4656 |
| 0.2 | 73.70 | 55.14 | 18.29 | 0.0144 | 0.6750 | 0.4385 |
| 0.1 | 82.84 | 55.41 | 18.35 | 0.0167 | 0.7355 | 0.4061 |
| 0 | 100.00 | 57.28 | 18.26 | 0.0185 | 0.7760 | 0.4090 |

where $k_{\text{used}}$ and $l_{\text{used}}$ are the cumulative API and latency costs consumed so far. We empirically set $\tau_0 = 0.2$, $K_{\max} = 0.02$, and $L_{\max} = 20$ based on preliminary tuning across all benchmarks. This configuration ensures that the router starts with a balanced offloading policy and becomes progressively more conservative as resources are consumed, maintaining overall cost efficiency without degrading reasoning quality.

**Summary.** Through this profiling-based training and normalization procedure, the router learns a unified utility function that generalizes across different task domains and cost settings. It enables HybridFlow to make consistent, cost-aware routing decisions that align closely with the optimization objective in Eq. 3.

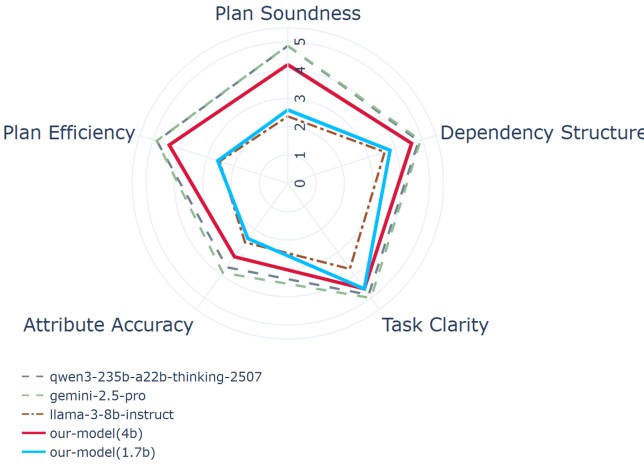

*Figure 5.* Results of our planner evaluation, assessing models on five key dimensions of task decomposition quality. We compare our two models (4B and 1.7B) against several leading standalone models, including Qwen3-235B-a22b-thinking-2507, Gemini-2.5-Pro, and Llama-3-8B-Instruct.

# D. Supplementary Experiments

To formally assess the quality of a generated plan, we introduce a dual-faceted evaluation framework that moves beyond singular metrics like final task accuracy. A superior Planner's quality is a function of both the intrinsic soundness of its generated plan and the extrinsic success of the Executor models in executing that plan. The absence of such a comprehensive metric in prior work makes objective comparison and targeted improvement of planning capabilities difficult. Our framework is designed to quantitatively measure these two facets, serving as the foundation for our subsequent data curation and model training efforts.

First, we assess the Intrinsic Plan Quality, which evaluates the machine-executable plan itself. This is judged across five key dimensions:

- **Plan Soundness & Decomposition:** This metric assesses whether the plan correctly and logically breaks down the problem. A flawed decomposition invalidates the entire solution strategy and is thus a primary point of evaluation.

- **Dependency Structure & Flow:** We evaluate the correctness of the task dependency graph. A logical dependency flow is crucial for maximizing parallelism and ensuring the correct context is passed between steps.

- **Task Clarity & Executability:** This dimension measures if each task is an unambiguous operational instruction suitable for an AI executor. Vague or poorly formulated tasks lead to poor downstream results.

- **Attribute Accuracy:** We judge the Planner's estimation of Difficulty and Token attributes. Accurate estimations are vital for the efficient dynamic allocation of models by the Router.

- **Plan Relevance & Efficiency:** This checks for redundant or irrelevant steps. A high-quality plan must be lean, purposeful and free of wasted computations.

Second, we measure the Extrinsic Execution Performance, which evaluates the efficacy of the execution models when acting upon the instructions provided by the Planner. This directly links plan quality to downstream performance and is assessed on the following five dimensions:

- **Instruction Following & Adherence:** This metric assesses how well the Executor model adheres to the specific constraints and instructions of the assigned sub-task.

- **Effective Use of Context:** We evaluate whether the model correctly utilizes the provided results from prior, dependent steps to inform its own execution.

- **Correctness & Factual Accuracy:** This measures the factual and logical accuracy of the model's response, serving as the primary measure of successful task completion.

- **Clarity & Machine Usability:** We judge whether the Executor's output is clear, well-structured, and easily parsable for use in subsequent steps.

- **Relevance & Conciseness:** This assesses if the model's response is concise and strictly relevant to the task, avoiding conversational filler or extraneous information.

Together, this dual-evaluation framework provides a comprehensive and structured methodology for analyzing both the plan and its real-world impact, enabling a systematic approach to enhancing the planning capabilities essential for effective hybrid model collaboration.

Recent advancements in enhancing Large Language Model (LLM) efficiency have increasingly focused on task decomposition. Prominent works such as SoT (Ning et al., 2024), DoT (Shao et al., 2025), and PASTA (Jin et al., 2025) exemplify this approach by leveraging parallel processing to improve time efficiency. However, these methods collectively raise two critical, unaddressed questions. First, **the field lacks a formal, quantitative methodology for measuring the intrinsic "quality" of the resulting plan itself**, making the comparison and optimization of planning capabilities difficult. Second, **it remains unknown whether this complex planning capability can be distilled from elite large models into a small model.**

We argue that addressing the second question is crucial for achieving efficient and scalable collaborative reasoning. If planning capabilities can be successfully distilled, a small model could undertake the core role of task decomposition. This would not only dramatically reduce latency and minimize API costs, yielding a cost-efficient, low-latency solution for complex reasoning, but it would also unlock new possibilities for edge deployment and even privacy-preserving scenarios (Li et al., 2024) where planning occurs locally. Therefore, the motivation for our work is twofold: first, through our Planner, we aim to establish a framework that can systematically evaluate plan quality; and second, we seek to demonstrate the feasibility of distilling this advanced capability from large to small models.

**Distillation of Planning Capabilities**  Recent studies have demonstrated that substantial performance gains can be achieved by fine-tuning models on small, high-quality datasets (Zhou et al., 2023; Muennighoff et al., 2025). This paradigm raises a critical research question for our work: **Can the sophisticated planning capabilities inherent in elite large models be effectively distilled into a small model using a similarly curated, high-quality dataset?**

This question is motivated by a significant performance gap observed in our own benchmark results (Table 1). We found that the intrinsic planning ability of small models, such as Llama-3-8B (Meta AI, 2024), is substantially inferior to that of state-of-the-art large models like GPT-5 (OpenAI, 2025). Bridging this gap is crucial for creating efficient and scalable collaborative systems.

To create this dataset, we developed a meticulous, benchmark-driven curation process. First, we used our Planner Evaluation Metric to identify top-performing LLMs. We then curated a set of "good" and "flawed" plan exemplars from their outputs to serve as didactic examples in a sophisticated meta-prompt. This prompt was used to guide a top-tier generative model to produce high-quality plans for the problems in the s1k dataset (Muennighoff et al., 2025).

We first evaluate the **effectiveness of the Planner**. This includes two aspects: the improvement in planning quality brought by Supervised Fine-Tuning (SFT), and the parallelization advantages introduced by task decomposition.

*Table 10.* Planner Performance Comparison
Worker: Llama3.2-3B, Dataset:GPQA

| Planner | Avg. Steps | $R_{\mathrm{comp}}$ | $C_{\mathrm{time}}$ | Acc |
|---|---|---|---|---|
| Llama3.2-3B base | 5.84 | 10.71 | 10.81 | 20.00 |
| Llama3.2-3B SFT | 6.12 | 34.3 | 11.59 | 22.00 |

**Effectiveness of SFT:** As shown in Table 10, our SFT Planner achieves an accuracy of 22.00% on GPQA, outperforming the planner based on the Llama3.2-3B base model (20.00% accuracy). This demonstrates the effectiveness of our distillation and fine-tuning process in generating high-quality, logically sound plans.

**Parallelization Advantage:** At the same time, Table 10 shows the SFT Planner achieves a 34.3% compression ratio. The compression ratio $R_{\mathrm{comp}}$ is defined as follows:

$$R_{comp} = (n - L_{crit})/n, \tag{28}$$

where $n$ is the total number of steps and $L_{crit}$ is the critical path length. This indicates that a significant number of steps within the tasks can be parallelized. HybridFlow's DAG decomposition aims to strike a balance between two extremes: **fully sequential execution** ($R_{comp} = 0$) and **fully parallel execution** ($R_{comp} = (n-1)/n$). Our method ensures accuracy by preserving critical dependencies while leveraging parallelization to significantly reduce end-to-end latency ($C_{time}$), making it faster than purely sequential execution.

### D.1. Privacy Discussion: Data Exposure in Edge–Cloud Collaboration

HybridFlow is designed for efficient inference under resource budgets; it is not a privacy-preserving protocol and does not provide formal privacy guarantees. Here we give a concise analysis of *data exposure* to the cloud, i.e., what information is transmitted in API requests.

**Threat model.**  We assume a standard cloud API setting where the cloud provider (or any entity with access to cloud logs) can observe the full content of each API request. We do not model a compromised device or side channels. The goal is to characterize and compare the amount of transmitted information across paradigms.

**Transmitted content under different paradigms.**  Let $Q$ denote the original user query. HybridFlow decomposes $Q$ into subtasks organized as a DAG. For a subtask $s_i$, let $\mathrm{Dep}(i)$ be its prerequisite subtasks and let $a_j$ be the generated answer for $s_j$. When HybridFlow offloads $s_i$ to the cloud, the transmitted payload contains:

$$x_i \triangleq \big(s_i, \ \{a_j\}_{j\in\mathrm{Dep}(i)}\big), \tag{29}$$

i.e., the current subtask and the answers of its dependencies, **without transmitting the original query** $Q$. In contrast, cloud-only inference transmits the full $Q$ (often along with additional prompts/history), while edge-only inference transmits nothing to the cloud.

**Exposure proxy.** To quantify cloud exposure in a model-agnostic way, we define a token-based proxy:

$$E_{\text{cloud}} \triangleq \sum_{i \in \mathcal{C}} \text{tok}(x_i), \tag{30}$$

where $\mathcal{C}$ is the set of subtasks routed to the cloud and $\text{tok}(\cdot)$ counts the number of tokens in the transmitted API payload.[2] We also report a normalized proxy

$$\bar{E}_{\text{cloud}} \triangleq \frac{E_{\text{cloud}}}{\sum_{i \in \mathcal{E}} \text{tok}(x_i) + \sum_{i \in \mathcal{C}} \text{tok}(x_i)}, \tag{31}$$

where $\mathcal{E}$ is the set of subtasks executed on the edge. This normalization reflects the fraction of total subtask-level information that is transmitted to the cloud.

**Implications and limitations.** HybridFlow can reduce exposure relative to cloud-only inference because it (i) offloads only a subset of subtasks and (ii) transmits only $(s_i, \{a_j\}_{j \in \text{Dep}(i)})$ rather than the full $Q$. However, exposure is not eliminated: if sensitive information is required to solve an offloaded subtask or is present in dependency answers $\{a_j\}$, it will be included in $x_i$ and observed by the cloud. More broadly, HybridFlow should be viewed as reducing the *surface area* of cloud-visible content in favorable cases, rather than guaranteeing privacy.

---

[2]If desired, one can include the cloud-generated output tokens as well by adding $\sum_{i \in \mathcal{C}} \text{tok}(y_i)$, where $y_i$ is the cloud response. We focus on transmitted inputs since they directly encode user-provided and intermediate information.

**Prompt:**

*You are a precise planning agent.*
Decompose the user's task into a sequence of concrete, easy-to-solve `sub_problems`.
Use high-level EAG-style roles implicitly (`Explain` → `Analyze` → `Generate`), but keep each `sub_problem` as a single sentence question.

*Plan Structure: EAG framework.*
`Explain`: To assist the following agents, what is your understanding of the question after reviewing it, focusing only on essential information and filtering out all irrelevant details.
`Analyze`: Break down the problem into the smallest possible, independent `sub_tasks` to solve the problem. These steps should rely on the "Explain" step or other completed analysis steps.
`Generate`: After reviewing the original question and the thoughts of previous agents, generate the final answer to the question.

*XML Plan Constraints*:
`id`: A unique integer (must be < 7 steps).
`task`: The question for executor AI.
`rely`: ID(s) of prerequisite steps (comma-separated if multiple).
Return ONLY the XML plan as final output. No additional text.

**Examples:**
```
<Plan>        <Step ID="1" Task="Explain:  What is the set (real numbers) and the
operation (multiplication) in question, and what is the core assertion (that it's
not a group) that needs to be verified?" Rely=""/>      <Step ID="2" Task="Analyze:
Check the closure property:  Is multiplication a binary operation on the set of
all real numbers?" Rely="1"/>   <Step ID="3" Task="Analyze:  Check the associative
property:  Is multiplication of real numbers associative?" Rely="1"/>  <Step ID="4"
Task="Analyze:  Check the identity property:  Is there an identity element for
multiplication in the set of real numbers?" Rely="1"/>   <Step ID="5" Task="Analyze:
Check the inverse property:  Does every element in the set of real numbers have a
multiplicative inverse?" Rely="1"/>   <Step ID="6" Task="Generate:  After reviewing
the original question and the thoughts of previous agents, what is the final
answer to the question?" Rely="2,3,4,5"/></Plan>

<Plan>   <Step ID="1" Task="Explain:  What is the base field, what are the adjoined
elements (sqrt(2), sqrt(3), sqrt(18)), and what is the required final output
format?" Rely=""/>      <Step ID="2" Task="Analyze:  What is the minimal polynomial
for sqrt(2) over Q, and what is the degree [Q(sqrt(2)) :  Q]?" Rely="1"/>        ...
(truncated for brevity) ... <Step ID="8" Task="Generate:  Based on the final degree
calculated in Step 7, what is the correct option letter and its corresponding
content?" Rely="1,7"/></Plan>

<Plan>   <Step ID="1" Task="Explain:  What is the set (real numbers), the operation
(multiplication), and the required output (option letter and content)?" Rely=""/>
...  (truncated for brevity) ...         <Step ID="6" Task="Generate:  Based on the
analysis of the group axioms in steps 2-5, which option correctly identifies
the reason this is not a group, and what is the final option letter and content?"
Rely="2,3,4,5"/></Plan>

Now is your turn:
```

*Figure 6. Planner prompt for task decomposition.*

**Original Question**:

"Given a reaction starting with 4,5-dimethylcyclohex-4-ene-1,2-dicarbonitrile and forming
2,3-dimethylbuta-1,3-diene and maleonitrile. Explain why adding maleic anhydride causes the
reaction to shift more to the right.
A. Maleic anhydride converts into 4,5-dimethylcyclohex-4-ene-1,2-dicarbonitrile and this causes
the equilibrium to shift towards reactants.
B. Maleic anhydride reacts with 4,5-dimethylcyclohex-4-ene-1,2-dicarbonitrile and addition of
a reactant causes the equilibrium to shift towards products.
C. Maleic anhydride reacts with 2,3-dimethylbuta-1,3-diene and removal of a product causes
the equilibrium to shift towards products.
D. Maleic anhydride converts into maleonitrile and addition of a product causes the
equilibrium to shift towards products."

**Task Decomposition and Execution**:

```
"subtasks": [{
    "id": "1",
    "task": "Explain: list the reactants, products, and the effect of adding maleic anhydride on the
    reaction.",
    "rely": [],
    "model_type": "cloud",
    "answer": "Reactants: 4,5-dimethylcyclohex-4-ene-1,2-dicarbonitrile \nProducts: 2,3-dimethylbuta
    -1,3-diene and maleonitrile  \nEffect of adding maleic anhydride: It reacts with one of the
    products, removing it from the equilibrium and shifting the reaction towards the products side."
},{
    "id": "2",
    "task": "Analyze: describe the reaction mechanism and equilibrium shift caused by maleic anhydride.",
    "rely": ["1"],
    "model_type": "edge",
    "answer": "Sub-problem 2 answer: \nMaleic anhydride reacts with 2,3-
    dimethylbuta-1,3-diene,
    forming maleonitrile and a dienol. The removal of maleonitrile from the equilibrium causes
    the reaction to shift more towards the product 2,3-dimethylbuta-1,3-diene, resulting
    in an increased yield of the desired product."
},{
    "id": "3",
    "task": "Analyze: evaluate the effect of maleic anhydride conversion to maleonitrile on the
    reaction equilibrium.",
    "rely": ["1", "2"],
    "model_type": "cloud",
    "answer": "The conversion of maleic anhydride to maleonitrile
    effectively removes maleonitrile, a product of the original reaction, from the equilibrium mixture.
    According to Le Chatelier's principle, removing a product shifts the equilibrium towards
    the products side to compensate. Therefore, this conversion drives the reaction forward,
    increasing the formation of 2,3-dimethylbuta-1,3-diene and maleonitrile.\n\n
    Final answer: C. Maleic anhydride reacts with 2,3-dimethylbuta-1,3-diene and removal
    of a product causes the equilibrium to shift towards products."
},{
    "id": "4",
    "task": "Generate: select the correct answer based on the analysis.",
    "rely": ["1", "2", "3"],
    "model_type": "edge",
    "answer": "Based on the analysis, the correct answer is:\n\nC. Maleic
    anhydride reacts with
    2,3-dimethylbuta-1,3-diene and removal of a product causes the equilibrium to shift
    towards products.\n\nThis is because maleic anhydride reacts with 2,3-dimethylbuta-1,3-diene to
    form maleonitrile and a dienol, and the removal of maleonitrile from the equilibrium causes the
    reaction to shift more towards the product 2,3-dimethylbuta-1,3-diene."
}]
```

**Final Answer**:

"C. Maleic anhydride reacts with 2,3-dimethylbuta-1,3-diene and removal of a product causes the equilibrium
to shift towards products."

*Figure 7.* HybridFlow case study.

