# OpenReview forum: "HybridFlow: Resource-Adaptive Subtask Routing for Efficient Edge-Cloud LLM Inference"
_ICML.cc/2026/Conference — ICML 2026 regular_

### Official Review · Reviewer_DPgq · 2026-03-07

**Soundness:** 3
**Presentation:** 2
**Significance:** 3
**Originality:** 3
**Overall Recommendation:** 4
**Confidence:** 3

**Summary:**

This paper proposes HybridFlow, an edge-cloud collaborative inference framework for LLM reasoning. The core idea is to decompose a complex query into a dependency-aware DAG of subtasks, and then dynamically route each ready subtask to either an edge model or a cloud model based on utility prediction and budget-adaptive thresholding. The paper claims that this design achieves a better balance among accuracy, latency, and API cost, and evaluates the method on GPQA, MMLU-Pro, AIME24, and LiveBench-Reasoning.

**Compliance With Llm Reviewing Policy:**

Affirmed.

**Final Justification:**

Thanks for the replies to my comments. I raised my score accordingly.

**Key Questions For Authors:**

1. Is it the DAG decomposition itself, the budget-aware routing formulation, or the online calibration under changing resource conditions? At present, these contributions are bundled together and the novelty boundary is not sufficiently sharp.
2. Were the DoT and PASTA baselines implemented in a way that faithfully reflects the original papers?
3. What is the exact source, preprocessing protocol, and evaluation setup for AIME24? The paper does not provide a sufficiently clear citation or benchmark description for this dataset.

**Strengths And Weaknesses:**

Strengths:
Edge-cloud collaborative reasoning for LLMs does require explicit trade-offs among accuracy, latency, and monetary cost, and the paper addresses a practical and timely systems problem.

Weaknesses:
1. The method description is not fully self-consistent. In the main text, the threshold is updated via the dual variable, i.e., 𝜏𝑡=clip(𝜏0+𝛾𝜆𝑡,0,1) while the appendix explains Eq. (11) using a different form that depends directly on used edge/cloud budgets. In addition, although the paper emphasizes online calibration via contextual bandits, the experiments do not isolate its contribution clearly. The ablations do not separate the effects of DAG scheduling, offline utility prediction, adaptive thresholding, and online calibration, so it remains unclear which component is primarily responsible for the reported gains.
2. From Tables 1 and 2, HybridFlow appears competitive among collaborative baselines, but it does not clearly dominate the strongest alternatives across the board. In particular, the highest average accuracy is still achieved by CoT + GPT-4.1, while HybridFlow’s strength seems to be a better accuracy-efficiency trade-off under lower API cost, rather than universal superiority. Some of the paper’s claims are stronger than what the results support.
3. PAPILLON appears twice in the references

---

> ### Author Rebuttal · Authors · 2026-03-30
>
> Sincerely thank you for the careful feedback. Legend: `W` = weakness, `Q` = question.
>
>
> **[Q1&W1a] Contribution split and what drives the gains.**
>
> Thank you for your feedback. We agree that the contributions should be separated more clearly. Our method has three components:
> 1. **Validated EAG-based DAG decomposition** with repair/fallback for robust execution;
> 2. **Utility-based routing over ready subtasks**, formulated through a **knapsack / primal-dual** view, so routing becomes a principled **accuracy–latency–cost allocation** problem rather than a heuristic rule;
> 3. **Contextual-bandit calibration** as an online correction layer beyond the offline profile.
>
> This split also clarifies the source of the gains. The **repair/fallback statistics** support the robustness of the decomposition layer. The budget-matched `Random` baseline shows that DAG structure alone is insufficient. The gap between `HybridFlow-Chain` and `HybridFlow` supports the benefit of dependency-aware DAG execution over a chain-style variant. The `Fixed Threshold` baseline shows that static routing is weaker than adaptive utility-based routing. Finally, `HybridFlow-NC` versus `HybridFlow` isolates the additional gain from contextual-bandit calibration.
>
> Overall, the improvement comes from the **combination** of robust decomposition, principled routing, and online calibration, rather than any single component in isolation. Among these, the largest qualitative gain comes from moving beyond heuristic or static routing to a **resource-aware routing formulation**, and calibration provides a further refinement. We will revise the paper to make this contribution split and the role of each ablation more explicit.
>
> |Setting|DAG Structure|Adaptive Threshold|Router|Calibration|GPQA Acc.|
> |---|---|---|---|---|---|
> |Random|Yes|No|No|No|46%|
> |HybridFlow-Chain|No|Yes|Yes|Yes|50.62%|
> |Fixed Threshold($\tau=0.5$)|Yes|No|Yes|Yes|51.62%|
> |HybridFlow-NC|Yes|Yes|Yes|No|51.85%|
> |HybridFlow|Yes|Yes|Yes|Yes|**53.33%**|
>
> **[W1b] Threshold semantics and analysis.**
>
> Thank you for your feedback. We want to point out the threshold update in the main text and the appendix is the **same mechanism described at two levels**. The main text gives the **theoretical view**, where a dual variable tracks resource pressure and maps it to $\tau_t$. The appendix gives the **implementation view**, instantiating that same pressure using cumulative API and latency usage, $(K_{\text{used}}, L_{\text{used}})$. Thus, these are not different update rules, but two views of the same control signal.
>
> More importantly, our paper already shows a **smooth threshold sweep on GPQA** (Figure 4, Table 6): for example, $\tau_0 = 0.5$ gives 51.62\% accuracy at \\$0.0088. This confirms that the threshold provides a controllable accuracy–cost trade-off.
>
> Our ablations further support the **adaptive router + threshold** design: at similar offload rates, it achieves a better accuracy–cost trade-off than non-adaptive alternatives. In addition, Figure 3 shows a meaningful **position-dependent offload pattern**: cloud usage concentrates on early, high-impact subtasks and tapers off as the budget tightens, which provides practical evidence that the adaptive threshold allocates resources sensibly over the reasoning process.
>
> **[W2] Claim scope relative to full-cloud.**
>
> Thank you for this helpful feedback. We agree that the strongest claim supported by our results is a better **accuracy–efficiency frontier among collaborative/hybrid baselines**, rather than universal superiority over unrestricted full-cloud inference.
>
> In particular, full-cloud CoT + GPT-4.1 remains the highest raw-accuracy setting. However, it also incurs substantially higher cost and latency: about **3.1×** HybridFlow’s API cost (**0.0269** vs. **0.0088**) and about **1.8×** its latency (**31.02s** vs. **17.48s**). Our method is therefore better understood from a **utility perspective**—that is, in terms of the **benefit–cost trade-off**—rather than raw accuracy alone.
>
> We will revise the wording accordingly and keep the claim narrow: HybridFlow is competitive in accuracy while providing a materially better cost–latency trade-off among hybrid methods.
>
> **[Q2] Reproduction of DoT and PASTA**
>
> Sure. DoT and PASTA were reproduced as closely as possible to their original protocols, specifically DoT with their source code[1]. They were then re-evaluated under the same shared edge/cloud model-pair setting used by HybridFlow so that differences come from the inference framework.
>
> [1]https://github.com/tsinghua-fib-lab/DoT
>
> **[Q3] AIME24 clarification** Thank you for catching the AIME24 description problem. AIME24 are math reasoning problems[2] and we only give LLM the problem descriptions and evaluate exact-match accuracy on the final integer answer.
>
> [2]https://huggingface.co/datasets/HuggingFaceH4/aime_2024
>
>
> **[W3] Bibliography issue** Thank you for catching the duplicate PAPILLON entry and we will fix it in the revision.

---

> > ### Author Rebuttal · Reviewer_DPgq · 2026-04-03
> >
> > Thanks for the replies to my comments. I raised my score accordingly.

---

> > > ### Author Response · Authors · 2026-04-03
> > >
> > > Thank you very much for confirming that we have fully addressed your concerns and considering raising your score. We truly appreciate your careful review and constructive feedback, which helped us clarify several important points.
> > >
> > > If you have any remaining concerns, we would be happy to clarify them further. If you find that your concerns have been fully resolved, we would sincerely appreciate your consideration of a higher rating. Thank you!

---

### Official Review · Reviewer_k3wp · 2026-03-09

**Soundness:** 3
**Presentation:** 3
**Significance:** 2
**Originality:** 2
**Overall Recommendation:** 4
**Confidence:** 3

**Summary:**

The paper proposes an edge-cloud hybrid system called HybridFlow for LLM inference. An inference task is first split into a DAG of subtasks, which are then dispatched to edge and cloud instances, respectively. HybridFlow identifies subtask dependencies and enables parallel execution of subtasks when possible. HybridFlow also features a utility model for deciding where to forward each subtask based on the trade-off between accuracy, cloud cost, and latency. Experimental results on multiple benchmarks show that HybridFlow reduces latency and cost while achieving similar accuracy.

**Compliance With Llm Reviewing Policy:**

Affirmed.

**Final Justification:**

I have raised my score to weak accept.

**Key Questions For Authors:**

- How does HybridFlow obtain real-time information about bandwidth variations and budget at runtime? How does the system adapt to changes in such factors in a fast manner?
- In Eq. (3), the accuracy achieved for different sub-tasks is summed up. Why do you think this is the correct way to model system accuracy, and how do you account for the accuracy dependency between the sub-tasks?
- It seems that for every task, a lot of profiling is needed. Can you provide some quantification regarding the profiling overhead? When is profiling needed?

**Limitations:**

Limitations are not directly mentioned in the paper, but some experiments and discussions are provided in the appendix. Consider moving some of the discussion points to the paper body.

**Strengths And Weaknesses:**

### Strengths

- The problem targeted by the paper is well motivated. The paper is well written in general.
- The evaluation is done thoroughly on several benchmarks, with comparisons to different methods representing the state of the art.

### Weaknesses

- The novelty of the paper is limited, given that the hybrid deployment model and the techniques applied have already been studied in HybridLLM, among others. The paper makes incremental extensions to existing work and does not seem to offer new insights into the problem or solution space.
- Some details are not clearly stated, particularly how the system “adapts” to factors like fast-changing network conditions and cost budgets. The experimental setup does not seem to reveal such important details, either.
- Subtask parallelism is claimed as one of the innovative ideas in the paper. However, I found this idea rather trivial. The idea is mainly speculative and depends heavily on the DAG structure and the routing decision. It would be interesting to dive deeper into this aspect and show its generalizability.
- The profiling overhead in HybridFlow seems quite high. A lot of profiling has to be done to train estimators for accuracy and latency approximation, as well as routing decision-making.

---

> ### Author Rebuttal · Authors · 2026-03-30
>
> Sincerely thank you for the careful feedback. Legend: `W` = weakness, `Q` = question.
>
> **[W1] Novelty boundary relative to prior hybrid work**
>
> We agree the novelty should be stated more sharply. It is not hybrid deployment alone, but the combination of a knapsack/primal-dual routing view, which turns offloading into a principled benefit-cost allocation problem, and validated DAG execution for ready-subtask routing and dependency-aware parallelism. Relative to `HybridLLM`, routing is finer-grained task level. Relative to `DoT`, decomposition is coupled with online budget updates and parallel scheduling. `HybridFlow-Chain` shows this matters: removing DAG parallel scheduling drops `GPQA` from `53.33%` to `50.62%`. Please refer to `Reviewer DPgq [Q1]` for more details.
>
> |Method|Limitation/delta|
> |---|---|
> |HybridLLM|coarse task-level hybrid routing; no dependency-aware ready-subtask scheduler|
> |DoT|decomposes, but sequential and not budget-adaptive online|
> |HybridFlow|knapsack-guided ready-subtask routing + dependency-aware parallel execution + online budget-aware routing|
>
> **[W2&Q1] Runtime adaptation to latency and budget changes**
>
> HybridFlow adapts from observed runtime signals, not from a separate bandwidth predictor. After each finished subtask, it records realized latency and API usage, updates cumulative $l_{used}$ and $k_{used}$, and recomputes $\tau_t = clip(\tau_0 + k_{used}/(2K_{max}) + l_{used}/(2L_{max}), 0, 1)$ before dispatching the next ready nodes. The router scores each ready subtask by predicted utility, and the calibration layer applies a lightweight online score correction from observed feedback. The update is closed-form, so adaptation happens at the next scheduling step, without retraining.
>
> | Runtime signal | Effect on later routing |
> | --- | --- |
> | realized latency increases | $\tau_t$ rises -> less later offloading |
> | realized API usage increases | $\tau_t$ rises -> less later offloading |
>
> **[W3]Subtask Parallellism and Scope**
>
> We thank the reviewer for this helpful point. We agree that parallel execution of independent subtasks, by itself, is not the main novelty. Our intended contribution is the joint design of (i) task decomposition and DAG construction, (ii) knapsack-guided ready-subtask routing over this DAG, and (iii) dependency-aware parallel execution, which together make parallel execution possible and useful when the task structure admits independent branches.
>
> We also agree that the benefit of subtask parallelism is conditional on the decomposition structure and routing decisions. Our claim is therefore not that all tasks can be parallelized, but rather that, on scientific reasoning tasks, a meaningful subset of instances exhibits partial-order structure that can be exploited to reduce latency while maintaining accuracy.
>
> Furtherly, we conduct additional test on GPQA to analyze fallback cases. We find comparable accuracy but a modest latency gap due to their chain execution structure. Importantly, fallback cases account for only a small fraction of the total samples, so their impact on the overall DAG-based evaluation is limited.
>
> |GPQA|number|Accuracy|Latency|API cost|
> |---|---|---|---|---|
> |valid|341|53.5%|15.27s|0.00745|
> |repaired|63|52.4%|15.33s|0.00767|
> |fallback|44|52.8%|16.73s|0.00762|
>
> **[Q2] Meaning of Eq. (3) and dependency between subtasks** Eq. (3) does not say that query accuracy is literally the sum of subtask accuracies (actually subtasks do not have accuracy). We use it as a routing surrogate: at each step, the system asks whether offloading one ready subtask is worth extra latency/API cost. $\Delta q_i$ is the marginal query-level gain of offloading subtask `i`, estimated from paired executions where only that subtask's placement changes while the rest of the pipeline is fixed. The DAG preserves dependency order, and final correctness is evaluated only on the complete answer. Thus the sum in Eq. (3) is a practical score for ranking routing decisions under budget, not an independence assumption about final accuracy.
>
> **[W4&Q3] Offline profiling burden and when it is paid** We agree that profiling introduces a real systems cost. However, it is not performed for every online query. In our setting, profiling is a one-time offline step conducted on a held-out set for a given model pair and deployment setting. Re-profiling is only needed when the model pair or deployment configuration changes.
>
> Accordingly, our Llama/GPT profiling audit should be interpreted as evidence of a one-time offline cost, rather than online runtime overhead. We report the corresponding latency and API cost below. Since the router itself is lightweight, we believe this profiling cost remains affordable in practice.
>
> |Profile audit|Value|
> |---|---|
> |queries / subtasks|2000 / 12400|
> |decompose total / avg|3240.2s / 1.62s|
> |paired execution total / avg|28595.4s / 14.30s|
> |total offline / avg|31835.6s / 15.92s|
> |LLM API cost/ avg|~\\$6.8 / ~\\$3.4e-3|
> |calls per query|6.2|

---

> > ### Author Rebuttal · Reviewer_k3wp · 2026-04-01
> >
> > Thank you for the response. Many of my concerns have been resolved and I would be happy to raise my score by one step. I am still not fully convinced about the novelty. Also, the profiling overhead seems high considering that each deployment setting might have high variability in network bandwidth and budget conditions.

---

> > > ### Author Response · Authors · 2026-04-02
> > >
> > > Thank you again for the thoughtful follow-up and for considering raising your score. We are glad that many of the original concerns have been addressed. Below we furtherly clarify the following two important questions: **(1) what is genuinely novel in HybridFlow**, and **(2) whether the profiling burden remains practical when deployment settings change**.
> > >
> > > **[Novelty.]**
> > >
> > > Our contribution is **not** limited to hybrid deployment or parallelism alone. The key problem we address is **online routing over ready subtasks**: under the **current latency or API budget**, the system must decide **which subtask to send to the edge or cloud**. It must also **respect dependencies and exploiting parallelism when possible**. We start from the **knapsack/primal-dual routing** view, which turns offloading into **a principled benefit-cost allocation problem**. We believe existing research does not account for these factors.
> > >
> > > In this sense, the difference from prior hybrid methods is not only architectural, but also theoretical. **HybridLLM** [1] performs **coarse task-level routing**. **DoT** [2] performs decomposition but runs slowly due to its **sequential** execution. It also cannot adapt online because the allocation signal is determined before execution begins. In contrast, **HybridFlow** integrates **validated DAG decomposition**, **fine-grained routing over ready subtasks**, and **online budget-aware control**. Importantly, the router is not just a heuristic threshold: it is motivated by a **knapsack / primal-dual utility view**, so that each routing decision can be interpreted as a **benefit-cost allocation** over expected accuracy gain, latency, and API cost. We believe this mechanism is the main novelty of our method.
> > >
> > > This distinction is also supported empirically. On the main GPQA routing ablation, **HybridFlow** attains the **highest utility (0.7940)**. We view these results as evidence that the gain is not simply from “using the cloud more,” nor from adding parallelism in a trivial way, but from the **joint effect of fine-grained routing and dependency-aware scheduling**. To avoid overstating the point, we will revise the paper to sharpen the claim accordingly: **HybridFlow is state-of-the-art among the evaluated edge-cloud collaborative baselines**.
> > >
> > > |Method|Accuracy(\%)|Latency(s)|API Cost(e-3\$)|Utility($\uparrow$)|
> > > |---|---|---|---|---|
> > > |HybridLLM [1]|52.9|15.96|16|0.4571|
> > > |DoT [2]|50.54|15.79|7.8|0.6494|
> > > |Random|46.00|15.15|7.5|0.5922|
> > > |Fixed Threshold|51.62|15.88|8.8|0.6292|
> > > |HybridFlow-Chain|50.62|16.12|8.2|0.6095|
> > > |HybridFlow (Our)|53.33|15.24|7.5|**0.7940**|
> > >
> > > **[Profiling burden and generalizability.]**
> > >
> > > We agree that profiling must be practical, otherwise the framework would be much less compelling. We would like to highlight that our profiling is a **one-time offline cost per deployment regime**, **not** a per-query online overhead. In our profiling audit, this amounts to `2000` queries, `31835.6s` total offline time (`15.92s/query`), and about `$6.8` total LLM API cost (`$3.4e-3/query`), all incurred **offline**. During online inference, the system does **not** re-profile for each budget state or short-term latency fluctuation. Instead, after each completed subtask, it observes the **realized latency and API usage** and updates the next routing threshold immediately. In other words, **offline profiling** provides the prior utility estimates, while **online adaptation** handles runtime variation.
> > >
> > > For changed deployment settings, we recognize that the most rigorous setup would re-profile when the **model pair**, **provider/device**, or **price/latency scale** changes substantially. At the same time, our results suggest that the method is **not brittle** and retains meaningful transfer. In **Appendix D.2 / Table 8**, after swapping the model pair to `Qwen2.5-7B + DeepSeek-V3`, our HybridFlow remains the strongest hybrid baseline on GPQA. We view this as evidence that the **routing design itself has useful cross-pair generalizability**: even when the edge/cloud pair changes, the framework still maintains a strong **accuracy–cost–latency trade-off**. So our position is intentionally modest: **full re-profiling is the rigorous option for a new deployment regime**, but the method already shows **useful transfer** rather than needing to be rebuilt from scratch whenever the setting changes.
> > >
> > > |`Qwen2.5-7B + DeepSeek-V3`|Accuracy(\%)|Latency(s)|API Cost(e-3\$)|
> > > |---|---|---|---|
> > > |HybridLLM [1]|47|47.87|3.63|
> > > |DoT [2]|49|40.90|1.80|
> > > |HybridFlow (Our)|**53**|**36.86**|**1.16**|
> > >
> > > We will revise the paper to make both points clearer. The novelty lies in the **budget-aware ready-subtask knapsack routing over a validated DAG**. The profiling cost is a **practical offline setup cost**, not an online burden. And our method has useful **cross-pair generalizability**.
> > >
> > > [1] https://arxiv.org/abs/2404.14618
> > >
> > > [2] https://arxiv.org/abs/2502.04392

---

### Official Review · Reviewer_V8WL · 2026-03-11

**Soundness:** 2
**Presentation:** 2
**Significance:** 3
**Originality:** 2
**Overall Recommendation:** 4
**Confidence:** 4

**Summary:**

This paper proposes HybridFlow, a resource-adaptive edge–cloud collaborative inference framework for large language models (LLMs). The goal is to improve the efficiency of LLM inference on edge devices by dynamically distributing computation between on-device small models and cloud-based large models. Instead of routing entire tasks to either edge or cloud, HybridFlow decomposes tasks into interdependent subtasks represented as a dependency-aware DAG, allowing them to be executed in parallel across heterogeneous resources.

The framework further introduces a benefit–cost driven routing mechanism that dynamically selects the execution location (edge or cloud) for each subtask based on latency and cost considerations. The authors evaluate the approach on several tasks and show improvements in latency and cost efficiency compared with baseline strategies.

**Compliance With Llm Reviewing Policy:**

Affirmed.

**Final Justification:**

The authors have addressed my comments. I raise my score.

**Key Questions For Authors:**

Subtask decomposition methodology
How are tasks automatically decomposed into subtasks?
Is the decomposition manually designed, rule-based, or generated through an automated mechanism?

Generalizability of the DAG representation
Can the proposed DAG-based representation be applied to arbitrary LLM tasks, or is it limited to specific types of structured tasks?

1. Overhead of DAG construction and scheduling.
What is the computational overhead introduced by dependency analysis and DAG scheduling?

2. Comparison with existing edge–cloud inference approaches.
How does HybridFlow differ fundamentally from recent collaborative inference frameworks that also distribute LLM workloads across heterogeneous resources?

3. Impact of subtask granularity.
How sensitive is the system performance to the number and granularity of subtasks?

4. Real-world deployment feasibility.
How would the proposed system operate in dynamic environments where network latency and cloud API performance fluctuate?

**Limitations:**

The current work has several limitations that should be addressed:

1. Unclear task decomposition process, making it difficult to reproduce the framework and evaluate its applicability across tasks.

2. Insufficient discussion of limitations in prior work, which weakens the motivation and novelty positioning of the proposed approach.

3. Limited analysis of system overhead, particularly related to DAG construction and scheduling.

4. Limited scalability analysis, especially under more complex task graphs or multi-user environments.

5. Insufficient ablation studies to understand the contribution of each design component.

**Strengths And Weaknesses:**

Strengths

1. Relevant and timely research problem. Edge–cloud collaborative inference for LLMs is an important emerging topic, especially for resource-constrained environments. Reducing inference latency and cost while maintaining acceptable performance is a highly relevant problem.

2. Interesting framework design. The idea of representing task execution as a dependency-aware DAG of subtasks and enabling parallel execution across edge and cloud resources is conceptually appealing and potentially useful for improving inference efficiency.

3. Resource-aware routing mechanism. The proposed benefit–cost based routing strategy attempts to balance latency and cloud API costs, which is an important practical consideration for real-world deployment.

4. Experimental evaluation. The paper includes empirical evaluations demonstrating latency and cost improvements compared to several baseline approaches.

Weaknesses

1. Unclear methodology for subtask decomposition (major clarity issue). A key component of HybridFlow is the decomposition of tasks into subtasks represented as a DAG, which directly determines how computation is distributed between edge and cloud. However, the paper does not clearly explain  how tasks are systematically divided into subtasks; whether this decomposition is manual, heuristic-based, or automatically generated; how the dependency relationships between subtasks are identified; whether the decomposition process generalizes across different task types. Since the effectiveness of the entire framework depends on the quality of this decomposition, the lack of clarity here significantly limits the reproducibility and generalizability of the approach.

2. Insufficient discussion of limitations in existing work. The related work section introduces prior research on edge–cloud inference and task routing but does not comprehensively analyze their limitations. For example, the paper claims that existing methods rely on coarse-grained task routing based on estimated task difficulty, but the discussion lacks.

3. Limited explanation of DAG construction overhead. The construction and management of the dependency-aware DAG may introduce additional overhead. However, the paper does not provide sufficient discussion or quantitative analysis of these overheads.

4. Limited analysis of scalability. The evaluation does not clearly demonstrate how the proposed approach scales when the number of subtasks increases. Scalability is particularly important for practical deployment in edge–cloud systems.

5. Limited ablation studies. The paper introduces multiple design components (e.g., DAG decomposition, benefit–cost routing, parallel execution), but the evaluation does not sufficiently isolate their individual contributions. Additional ablation studies would help clarify which components are responsible for the observed performance gains.

6. Evaluation scenarios could be more diverse. The experiments appear to be conducted on a limited set of workloads or tasks. It would strengthen the work to include more diverse scenarios.

---

> ### Author Rebuttal · Authors · 2026-03-27
>
> Sincerely thank you for the careful feedback. Legend: `W` = weakness, `Q` = question.
>
> **[W1&Q1] Automatic decomposition methodology**
>
> The decomposition is automatic, not manual. For each query, the same Explain-Analyze-Generate (EAG) planner prompt produces an XML plan; each `<Step/>` contains task text and `Rely`, and the parser converts them into DAG nodes and prerequisite edges. We then validate acyclicity, a unique `Explain` root, reachability, one final `Generate` sink, `n <= 7`, and dependency consistency. Minor violations are repaired by bounded rules; only persistent failures revert to a sequential chain. This decomposition is a general paradigm that adapts to all benchmarks in our paper.
>
> **[W2&Q4] Limitations of existing work and our difference**
>
> HybridFlow differs by combining ready-subtask routing, dependency-aware parallel scheduling, and online budget adaptation. In our comparison setting, `HybridLLM` is a coarser-grained hybrid baseline: it collaborates across edge/cloud, but does not route at dependency-aware subtask granularity or use the same DAG scheduler. `DoT` performs subproblem decomposition, but their implementation remains sequential, and its allocation signal is a pre-execution difficulty estimate rather than a threshold updated by cumulative latency/API usage. HybridFlow addresses these gaps jointly: every ready subtask is routed online, independent subtasks can run in parallel across edge/cloud, and routing tightens as budget is consumed within the same query.
>
> **[W3&Q3] DAG construction overhead**
>
> The overhead has two parts. The meaningful part is planner-model DAG generation; in a completed `GPQA`-128 proxy artifact it averages `2.14s/query` (min/max `1.95/2.55`) versus `15.4s` average end-to-end time, and that planner time is already counted in reported latency. Post-generation dependency analysis is much smaller: XML parsing averages `0.0115 ms` and DAG bookkeeping / ready-queue maintenance `0.0043 ms`. As planner time is included in reported latency, we therefore claim this affordable and reasonable overhead.
>
> |Stage|Cost|
> |---|---|
> |planner DAG generation|`2.14s` avg (`1.95-2.55s`)|
> |XML parse after planner output|`0.0115 ms` avg|
> |DAG bookkeeping after planner output|`0.0043 ms` avg|
> |note|already counted in end-to-end latency|
>
> **[W4&Q5] Subtask granularity.**
>
> Performance is indeed sensitive to subtask granularity, which is why we cap the planner at $n \leq 7$. If the decomposition is too coarse (small $n$), potential concurrency is lost; if it is too fine (large $n$), planning and synchronization overhead increase.
>
> To further examine this effect, we report a **GPQA-128 proxy audit** grouped by plan size. The results show that **1–4-task** plans and **5–7-task** plans achieve very similar accuracy and API cost, while the latter are only slightly slower, mainly due to their longer chain-style execution structure. Within the observed $\leq 7$-task regime, this suggests that the method is reasonably stable. We will clarify in the revision that our claim is stability within the **bounded granularity regime** we study.
>
> |GPQA-128 bin|Number|Acc.|Avg. Lat.|API Cost|
> |---|---|---|---|---|
> |`1-4` tasks|87|54.02%|14.77s|0.007546\\$|
> |`5-7` tasks|41|53.66%|16.72s|0.007623\\$|
>
> **[W5] Ablation study**
>
> We agree clearer component isolation helps. The submission already contains two targeted ablations. `HybridFlow-Chain` keeps routing but removes DAG parallel scheduling, dropping `GPQA` from `53.33%` to `50.62%`; this isolates the contribution of dependency-aware parallel execution. The budget-matched `Random` router uses the same `C_API = 0.0075` but reaches only `46.00%`, so the gain is not just more cloud usage.
>
> |Setting|GPQA Acc.|What it isolates|
> |---|---|---|
> |Random|46.00%|naive budget-matched routing|
> |HybridFlow-Chain |50.62%|no DAG parallelism|
> |Fixed Threshold($\tau=0.5$)|51.62%|no adaptive threshold|
> |HybridFlow-NC|51.85%|no calibration|
> |HybridFlow|**53.33%**|full system|
>
> **[Q2&W6&Q6] DAG scope and dynamic deployment.**
>
> Thank you for the feedback. We do not claim that arbitrary LLM tasks naturally admit a DAG representation; the intended scope is **decomposable multi-step reasoning tasks**. Accordingly, our baselines and experiments focus on LLM reasoning, and the current paper is specifically centered on **scientific reasoning tasks**. Extending transfer evaluation to broader domains, such as multimodal tasks, is an important direction for future work.
>
> For deployment, the current system does not rely on an explicit bandwidth predictor. Instead, the routing threshold is updated from **cumulative observed latency and API usage**, so slower or more expensive cloud calls automatically make later offloading more conservative within the same query. We therefore keep the deployment claim narrow: the method is **adaptive to observed runtime conditions**, but stronger claims about highly fluctuating networks would require dedicated experiments.

---

> > ### Author Rebuttal · Reviewer_V8WL · 2026-04-04
> >
> > Thank you for the reply, my comments have been addressed.  I will raise my score.

---

> > > ### Author Response · Authors · 2026-04-04
> > >
> > > Thank you very much for confirming that we have fully addressed your concerns and raising your score. We truly appreciate all the valuable advice we have received, and we are pleased to share that other two of the reviewers also have kindly recognized our improvements by raising their scores, making our paper higher than the accept score line (4 and above). This acknowledgment reflects the positive impact of our collaborative efforts in enhancing the quality of the paper.
> > >
> > > If you have any remaining concerns, we would be happy to clarify them further. If you find that your concerns have been fully resolved, we would sincerely appreciate your consideration of a higher rating. Thank you!

---

### Official Review · Reviewer_tnK5 · 2026-03-13

**Soundness:** 3
**Presentation:** 2
**Significance:** 3
**Originality:** 3
**Overall Recommendation:** 4
**Confidence:** 3

**Summary:**

The authors tackle a v. practical problem! Edge devices often can't run powerful enough LLMs locally, but sending everything to the cloud is slow and expensive. Existing edge-cloud routing systems make coarse, query-level decisions based on difficulty estimates, missing opportunities for parallelism and failing to adapt to changing budgets. HybridFlow addresses this with two main ideas ---

- A dependency-aware task decomposition step that breaks each query into a DAG of subtasks, allowing independent subtasks to run in parallel rather than forcing sequential execution
- A learned utility-based router that decides, for each subtask, whether to run it locally on the edge model or offload to the cloud — balancing accuracy gain against latency and API cost, with an adaptive threshold that tightens as the budget gets consumed

The routing mechanism is based on a knapsack optimization formulation, with the adaptive thresholding shown to be an instance of primal-dual updates on the Lagrangian relaxation.

The final result is a system that gets close to full-cloud accuracy while using substantially less API budget and achieving lower latency than sequential baselines.

**Compliance With Llm Reviewing Policy:**

Affirmed.

**Final Justification:**

The responses addressed my concerns and I would like to keep my positive score.

**Key Questions For Authors:**

1. **Is the contextual bandit calibration active in the main experiments?** Section 3.3 says it "can be enabled," but Section 4.1 describes it as part of the implementation. This ambiguity makes it hard to know what system the main results actually reflect. If it's on, the ablation table should include a version without it; if it's off, the description in 4.1 is misleading. Clarifying this is important for reproducibility and would affect my assessment of the presentation score.

2. **How sensitive are the results to the normalization constants in Eq. (24)?** The values of 10s for latency and $0.02 for API cost appear to be set by hand and are never ablated. A brief sensitivity analysis here would substantially strengthen the soundness of the routing mechanism

3. **How does the offline-profiled router transfer to new task domains?** The router is trained on MMLU-Pro and Math500, then evaluated on GPQA, AIME24, and LiveBench-Reasoning. To what extent does cross-domain transfer hold up? If re-profiling is needed for each new domain, the practical overhead is significant and should be quantified.

4. **How does HybridFlow perform against a budget-matched baseline?** The most natural soundness check for the routing mechanism is a baseline that uses the same total API budget as HybridFlow but allocates it naively (e.g., random or fixed-threshold routing). The random baseline in Table 3 gets close to this but uses a slightly different offload rate (42.1% vs. 40.48%). A cleaner budget-matched comparison would help isolate how much of the accuracy gain comes from smarter routing vs. the amount of cloud usage

**Limitations:**

Yes

**Strengths And Weaknesses:**

### Strengths

- **The knapsack framing is an interesting contribution.** Casting subtask routing as a 0-1 knapsack problem and showing that the adaptive thresholding mechanism is an instance of primal-dual updates on the Lagrangian relaxation is clean and gives the system principled theoretical grounding.

- **The combination of DAG-based parallelism with budget-aware routing is well-motivated and novel in this setting.** Prior work either parallelizes aggressively without respecting dependencies (SoT, PASTA) or routes adaptively but sequentially (DoT, HybridLLM). HybridFlow's contribution is explicitly coupling these two concerns, and the ablation (HybridFlow-Chain) does a good job isolating how much each piece contributes.

- **The ablation study is solid.** Table 3 shows clearly that routing alone without DAG parallelism (HybridFlow-Chain) is meaningfully worse, and that random routing at the same offload rate underperforms the learned router.

- **The model-pair swap experiment (Table 8) is infromative.** Showing that the system transfers to Qwen2.5-7B + DeepSeek-V3 without retraining meaningfully strengthens the generality claim

- **The position-dependent offload pattern (Figure 3) is an interesting finding.** The observation that cloud usage concentrates on early, high-impact subtasks and tapers off as the budget tightens is intuitive but not obvious, and it's a useful insight for practitioners

### Weaknesses

- **The accuracy gains over the best baselines are modest and the comparison is uneven.** HybridFlow achieves 55.34% average accuracy vs. CoT with GPT-4.1 at 58.99% — but CoT with GPT-4.1 uses the cloud for everything. The more meaningful comparison is against DoT (46.50%) and HybridLLM (38.70%), where HybridFlow does look better, but these baselines use the same Llama3.2-3B + GPT-4.1 pair. It would be useful to see a stronger hybrid baseline that uses the same total API budget as HybridFlow to isolate how much of the gain comes from smarter routing vs. simply making more cloud calls

- **The offline profiling requirement is a significant practical limitation that is underplayed.** The router is trained on 2,000 queries sampled from MMLU-Pro and Math500, with paired edge/cloud executions for every subtask. This is expensive and domain-specific. it's not clear how well the learned utilities transfer to a new task domain without re-profiling. The paper acknowledges this briefly and doesn't quantify cross-domain transfer degradation.

- **The planner reliability numbers (Table 5) raise concerns that aren't fully addressed.** Only 76-78% of plans are valid without repair, and 9-10% fall back to sequential chains. The paper reports this transparently, which is good, but doesn't analyze how fallback cases affect the overall accuracy and latency numbers. If fallbacks are systematically harder queries, the reported averages may be optimistic about the DAG execution path.

- **Presentation has some gaps** The contextual bandit calibration component (Section 3.3) is described at a high level but it's unclear whether it's actually used in the main experiments or only optionally enabled. Similarly, the normalization constants in Eq. (24) (10 seconds for latency, $0.02 for API cost) appear to be set by hand and their sensitivity is never evaluated

- **The significance is somewhat bounded by the experimental scope.** All experiments use a single RTX 3090 for edge computation and GPT-4.1 via API for cloud. Real edge deployments involve tighter memory constraints, intermittent connectivity, and much lower-end hardware. The paper doesn't discuss how the system behaves under network variability or memory pressure, which are central concerns for this setting

---

> ### Author Rebuttal · Authors · 2026-03-27
>
> Sincerely thank you for the careful feedback. Legend: `W` = weakness, `Q` = question.
>
> **[W1&Q4] Budget-matched hybrid comparison**
>
> We agree the fairest comparison is among hybrid methods under similar budgets. Table 3 already gives a near budget-matched GPQA check. Here we furtherly provide a budget-matched comparison with similar offload rate and api cost, yet HybridFlow is much more accurate. Therefore, the gain is not simply from making more cloud calls but our budget-aware router.
>
> |Method|Acc.|Cost|Offload Rate|
> |---|---|---|---|
> |Random|45.06%|0.0075|40.4%|
> |Fixed-threshold(tau=0.5)|50.89%|0.0078|40.6%|
> |HybridFlow|**53.33%**|0.0075|40.4%|
>
> **[W2&Q3] Domain/model transfer and profiling burden**
>
> For domain transfer, our router is profiled on held-out queries from MMLU-Pro and Math500, while the main evaluation is conducted on GPQA, AIME24, and LiveBench-Reasoning, reflecting transfer to out-of-profiling tasks, rather than evaluation on the same distribution used for profiling. However, we sincerely acknowledge that the current scope of our work is scientific reasoning tasks. Extending transfer evaluation to broader domains, such as multimodal tasks, is an important direction for future work.
>
> For model transfer, Table 8 in the appendix presents a model-pair swap experiment on GPQA, where we keep the overall pipeline fixed and replace the edge/cloud model pair with Qwen2.5-7B and DeepSeek-V3. Under this setting, HybridFlow still outperforms the other hybrid baselines, suggesting our method is not tightly coupled to a single model pair and retains its advantage under a different edge/cloud configuration.
>
> Accordingly, our Llama/GPT profiling audit should be interpreted as evidence of a one-time offline cost, rather than online runtime overhead. We report the corresponding latency and API cost below. Since the router itself is lightweight, we believe this profiling cost remains affordable in practice.
>
> |Profile audit|Value|
> |---|---|
> |queries / subtasks|2000 / 12400|
> |decompose total / avg|3240.2s / 1.62s|
> |paired execution total / avg|28595.4s / 14.30s|
> |total offline / avg|31835.6s / 15.92s|
> |LLM API cost/ avg|~\\$6.8 / ~\\$3.4e-3|
> |calls per query|6.2|
>
> **[W3] Repair and fallback cases**
>
> To analyze how fallback cases affect the overall accuracy and latency numbers, we conduct additional test on full GPQA. We find comparable accuracy in fallback cases, but a modest latency gap due to their chain execution structure. Importantly, fallback cases account for only a small fraction of the total samples, so their impact on the overall DAG-based evaluation is limited.
>
> |GPQA|number|Acc.|Latency|API cost|
> |---|---|---|---|---|
> |valid|341|53.5%|15.27s|0.00745|
> |repaired|63|52.4%|15.33s|0.00767|
> |fallback|44|52.8%|16.73s|0.00762|
>
>
> **[W4&Q1] Contextual-bandit calibration**
>
> Yes, the contextual-bandit calibration is active in HybridFlow and is an integral part of the method. We agree that this is not stated clearly enough in the current draft. To address this point, we provide an ablation comparing the offline-profiled router without calibration against the full HybridFlow system across benchmarks.
> The results support our conclusion that the calibration layer is beneficial. Since the router is offline-profiled on MMLU-Pro, the no-calibration variant still achieves comparable performance on the in-distribution benchmark. However, it shows a noticeable drop on the other benchmarks, where online calibration appears to be more important.
>
>
> |Setting|GPQA|MMLU-Pro|AIME24|Livebench-Reasoning|
> |---|---|---|---|---|
> |router without calibration|51.85%|71.82%|20%|54.48%|
> |full HybridFlow|**53.33%**|**72.54%**|**36.67%**|**58.83%**|
>
>
> **[W4&Q2] Normalization sensitivity**
>
> We agree Eq. (24) should be stress-tested rather than treated as a single brittle setting. To address this point, we provide a matched sweep on GPQA to test the nomalization sensitivity. The main signal is that the operating point is tunable rather than uniquely brittle.
>
> |Scale $(l\^{sub}\_{max}, k\^{sub}\_{max})$|Acc.|API Cost|Offload|
> |---|---|---|---|
> |(5, 0.02)|35.04%|0.0043|25.50%|
> |(10, 0.01)|37.95%|0.0046|27.22%|
> |(10, 0.02)default|**53.33%**|0.0075|40.48%|
> |(40, 0.04)|53.25%|0.0166|88.83%|
>
> **[W5] Deployment scope**
>
> We agree the evaluated setup is controlled: `Llama3.2-3B` on one RTX 3090 for edge-side planning and `GPT-4.1` via API for the cloud. Runtime adaptation uses observed cumulative latency and API usage, so slower or more expensive cloud use tightens the threshold within the query. We therefore keep the claim conservative: the current evidence supports a stronger accuracy-efficiency frontier among collaborative baselines, but not broader claims about intermittent connectivity, lower-end edge hardware, or severe memory pressure.

---

> > ### Author Rebuttal · Reviewer_tnK5 · 2026-04-03
> >
> > This resolved my questions. I will keep my score as is.

---

> > > ### Author Response · Authors · 2026-04-03
> > >
> > > Thank you very much for the follow-up and for confirming that your concerns have been fully resolved. We truly appreciate all the valuable advice we have received, and we are pleased to share that two of the reviewers have kindly recognized our improvements by raising their scores. This acknowledgment reflects the positive impact of our collaborative efforts in enhancing the quality of the paper.
> > >
> > > If there are any remaining questions, we would be very happy to clarify them further. If you find that your concerns have been resolved, we would sincerely appreciate your consideration of a higher rating. Thank you!

---

### Decision · Program_Chairs · 2026-04-30

**Decision:**

Accept (regular)

**Comment:**

The paper addresses an important practical problem in edge–cloud LLM inference and presents a well-motivated combination of dependency-aware task decomposition and budget-aware subtask routing. Its main contribution is not hybrid deployment alone, but the integration of validated DAG-based execution with online utility-driven routing, which leads to a stronger accuracy–latency–API cost trade-off than the collaborative baselines evaluated. The original reviews raised legitimate concerns about the clarity of the decomposition procedure, the role of the calibration component, the novelty relative to prior hybrid methods, and the practical burden of offline profiling. After the rebuttal and subsequent reviewer updates, these concerns were addressed sufficiently for acceptance: the authors clarified the automatic decomposition and repair pipeline, added ablations isolating key components, provided budget-matched comparisons, and quantified fallback behavior and profiling overhead. The remaining limitations are real but not decisive, i.e., the novelty is somewhat bounded and should be framed more carefully, the experimental setting is controlled rather than fully representative of harsher deployment conditions, and some claims in the draft should be sharpened to emphasize superiority over the evaluated collaborative baselines rather than over unrestricted full-cloud reasoning. Even so, the paper appears technically sound, empirically credible, and likely to be useful to the ICML community working on efficient LLM systems and edge–cloud collaboration, so I recommend acceptance.